# Molecular determinants of inhibition of UCP1-mediated respiratory uncoupling

Antoine Gagelin[1,2,5], Corentin Largeau[1,2,3,5], Sandrine Masscheleyn[2,3], Mathilde S. Piel[2,3], Daniel Calderón-Mora[2,3], Frédéric Bouillaud[4], Jérôme Hénin ®[1,2,6] ✉ & Bruno Miroux ®[2,3,6] ✉

Brown adipose tissue expresses uncoupling protein 1 (UCP1), which dissipates energy as heat, making it a target for treating metabolic disorders. Here, we investigate how purine nucleotides inhibit respiration uncoupling by UCP1. Our molecular simulations predict that GDP and GTP bind UCP1 in the common substrate binding site in an upright orientation, where the base moiety interacts with conserved residues R92 and E191. We identify a triplet of uncharged residues, F88/I187/W281, forming hydrophobic contacts with nucleotides. In yeast spheroplast respiration assays, both I187A and W281A mutants increase the fatty acid-induced uncoupling activity of UCP1 and partially suppress the inhibition of UCP1 activity by nucleotides. The F88A/I187A/ W281A triple mutant is overactivated by fatty acids even at high concentrations of purine nucleotides. In simulations, E191 and W281 interact with purine but not pyrimidine bases. These results provide a molecular understanding of the selective inhibition of UCP1 by purine nucleotides.

Uncoupling protein 1 (UCP1), a member of the SLC25 mitochondrial carrier family, is found in brown adipose tissue (BAT), an organ specialized in non-shivering thermogenesis in small mammals and some adult humans. UCP1 is abundant in the inner membrane of BAT mitochondria, where it dissipates the electrochemical proton gradient and thus uncouples respiration from ATP synthesis, which in turn increases respiration, reoxidation of coenzymes, substrate oxidation, and energy release as heat. UCP1 transports protons when activated by micromolar concentration of free fatty acids (FFAs) released by lipolysis of triglycerides[1].

Stimulating energy expenditure, as occurs in brown fat upon cold exposure, is an attractive approach for treating metabolic diseases such as obesity and diabetes. Adipocyte browning by expression of UCP1 has been successfully attempted in obese mice models and is a promising avenue for human therapy[2,3]. However, increasing UCP1 levels in adipocyte mitochondria may not be sufficient to significantly modify energy balance and body mass in humans because, in basal conditions, wild-type UCP1 is inhibited by millimolar

concentrations of purine nucleotides[1,4]. This work aims to get molecular insight into UCP1 regulation and to design unregulated variants exhibiting enhanced uncoupling activity and less inhibition by nucleotides.

There is no experimentally resolved structure of UCP1. However, a general mechanism of transport by mitochondrial carriers has been proposed based on atomic structures of the related ADP/ATP carrier (AAC)[5]. In the C-state conformation of AAC induced by carboxyatractyloside (CATR)[6,7], the cavity is widely open toward the cytoplasmic side, allowing ADP to enter the positively charged electrostatic funnel[8,9]. Within the common substrate binding site located in the middle of the cavity, ADP is stabilized by electrostatic interactions with positively charged amino acids. This charged motif is supported by a matrix salt bridge and a highly conserved glutamine brace network at the C-terminus of odd α-helices[10]. In the matrix-facing M-state induced by bongkrekic acid (BKA)[11], the orientation of the cone-shaped protein is inverted upon the formation of a cytoplasmic salt bridge network found at the C-terminus of

[1]Université Paris Cité, Laboratoire de Biochimie Théorique CNRS UPR9080, Paris 75005, France. [2]Institut de Biologie Physico-Chimique, Fondation Edmond de Rothschild, Paris 75005, France. [3]Université Paris Cité, Laboratoire de Biologie Physico-Chimique des Protéines Membranaires CNRS UMR7099, Paris 75005, France. [4]Université Paris Cité, Institut Cochin, Inserm U1016, CNRS UMR8104, Paris 75014, France. [5]These authors contributed equally: Antoine Gagelin, Corentin Largeau. [6]These authors jointly supervised this work: Jérôme Hénin, Bruno Miroux. ✉e-mail: jerome.henin@cnrs.fr; bruno.miroux@ibpc.fr

even helices. In the alternate access model, substrate binding drives the conformational changes required for the transition from C- to M-state, allowing a ping-pong-like mechanism of transport to occur[5]. Mitochondrial carriers exhibit an order-3 pseudo-symmetry due to an ancestral gene triplication[12]; as a result, their sequences can be analyzed based on symmetric or asymmetric sets of three paralogous residues (triplets) in the common substrate binding site[7,13]. Like those of all members of the family, UCP1 sequences show highly-conserved consensus sequences involved in both salt bridge networks that are critical for the alternate access transport mechanism[10]. However, it is still unclear how FFAs mediate proton transport in either UCP1 or AAC[14,15] and how nucleotides, which are transported by AAC but not by UCP1, inhibit UCP1 activity.

In this study, we build homology models of UCP1 in the two most likely putative conformations and perform molecular dynamics simulations with accelerated sampling methods to explore the accessibility and interactions of nucleotides within the UCP1 cavity. To experimentally confirm the predictions obtained from MD simulations, we express UCP1 wild-type and mutants in yeast and assess the respiratory response to activators and inhibitors. As a result, we identify unique molecular features of UCP1 that distinguish it from other mitochondrial carriers and find hyperactive UCP1 mutants that may be suitable for increasing energy expenditure in a therapeutic context.

## Results

### Structural models of UCP1

The structure with the closest sequence to UCP1 is one of UCP2 (58% identity), which was obtained by liquid-state NMR[16] in the presence of DPC. This detergent perturbs the function and structure of AAC and UCP1[17–21]. High-resolution crystal structures exist for the ATP/ADP Carrier (AAC), which has also been shown to exhibit proton transport activity and to contribute to fatty-acid-dependent uncoupling of respiration[15]. Recently, AAC structures have been used to model UCP2[22].

Here, homology models of rat UCP1 were built based on AAC crystal structures in the C-state[23] and M-state[11], and for comparison, the NMR structure of UCP2[16]. While multiple sequence alignments (MSAs) of a wider variety of homologous sequences were produced, they proved unnecessary for modeling due to the smaller MSA's lack of ambiguity. Figure 1a-b represents multiple sequence alignments between the three paralogous two-helix repeats within each sequence. This highlights major conserved features as well as a significant variation: helix 6 of the AAC sequence exhibits a deletion with respect to all other even-numbered helices at the position of UCP1 L278. The only case of manual adjustment of the alignments, at position L278, was not decided based on sequence information but rather on structural behavior in simulations (see Methods for details).

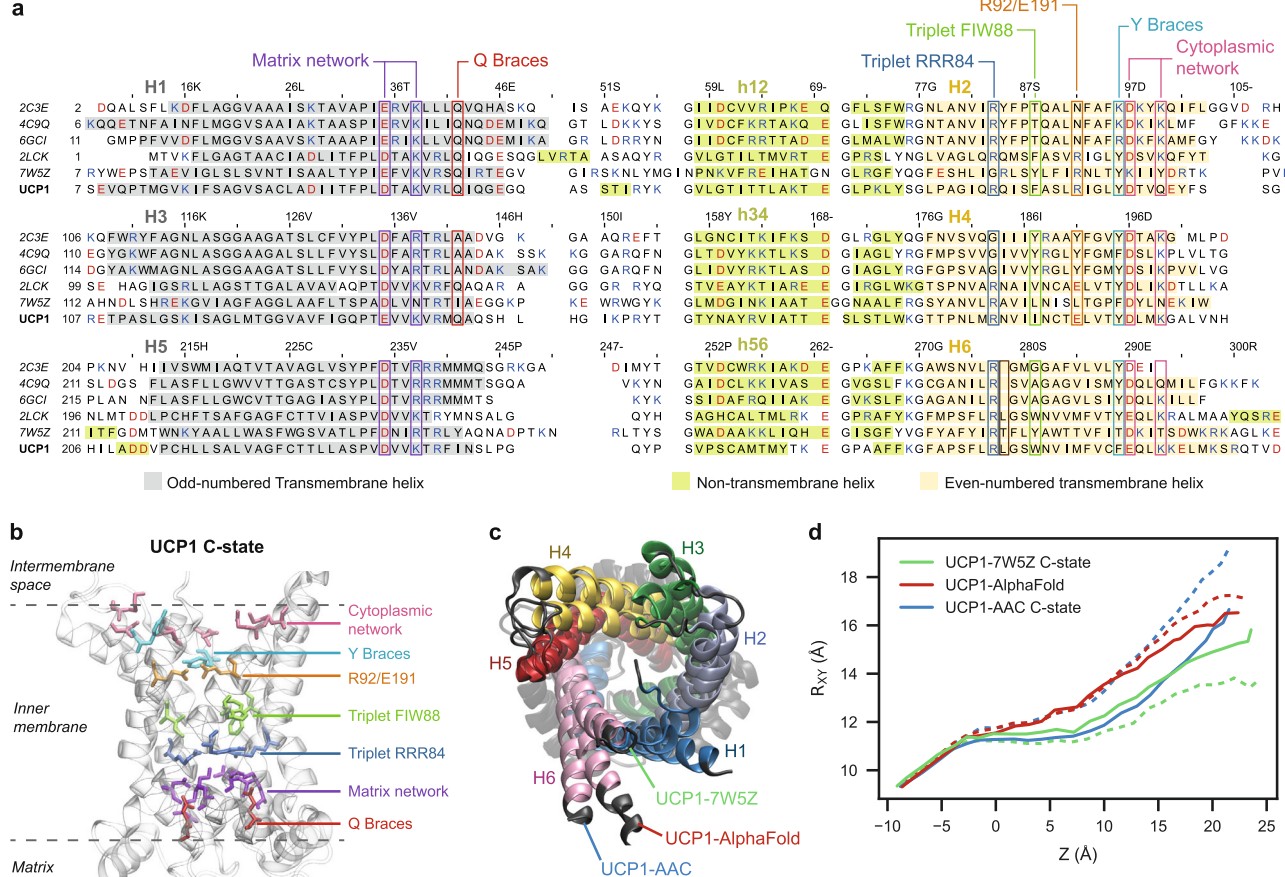

**Fig. 1 | UCP1 models tend toward a consensus apo C-state conformation. a** Multiple alignments of the sequences resolved in the structures of AAC (PDB 2C3E *Bos taurus*, PDB 4C9Q *Saccharomyces cerevisiae* and PDB 6GCI *Thermothelomyces thermophilus*), UCP2 (PDB 2LCK, *Mus musculus*), *Tetrahymena thermophila* putative 2-oxoglutarate/malate carrier protein (PDB 7W5Z) and *Rattus norvegicus* rUCP1. Residue numbers above the alignment refer to the UCP1 sequence. Colored backgrounds highlight α-helices detected in experimental structures of AAC and UCP2 and our C-state model of UCP1 (PDB 2C3E). H1 to H6 and h12 to h56 are the standard names for the helices of the mitochondrial carriers, as defined by Pebay-Peyroula, E. et al.[6]. **b** Cartoon representation of the UCP1 model based on AAC C-state (PDB 2C3E), viewed from the membrane plane. **c** Overlapped cartoon representations of three rUCP1 models: based on AAC C-state (PDB 2C3E), based on a putative C-state-like mitochondrial carrier in structure 7W5Z, and produced by AlphaFold. **d** Helix splay profiles along the transmembrane axis (Z axis) for three C-state-like models initially (dashed lines) and after 300 ns relaxation by all-atom simulation in a membrane (solid lines). UCP1-7W5Z C-state and UCP1-AAC C-state profiles are replica-averaged profiles. The number of replicas, $n = 3$, except for UCP1-Alphafold $n = 1$. Source data are provided as a Source Data file.

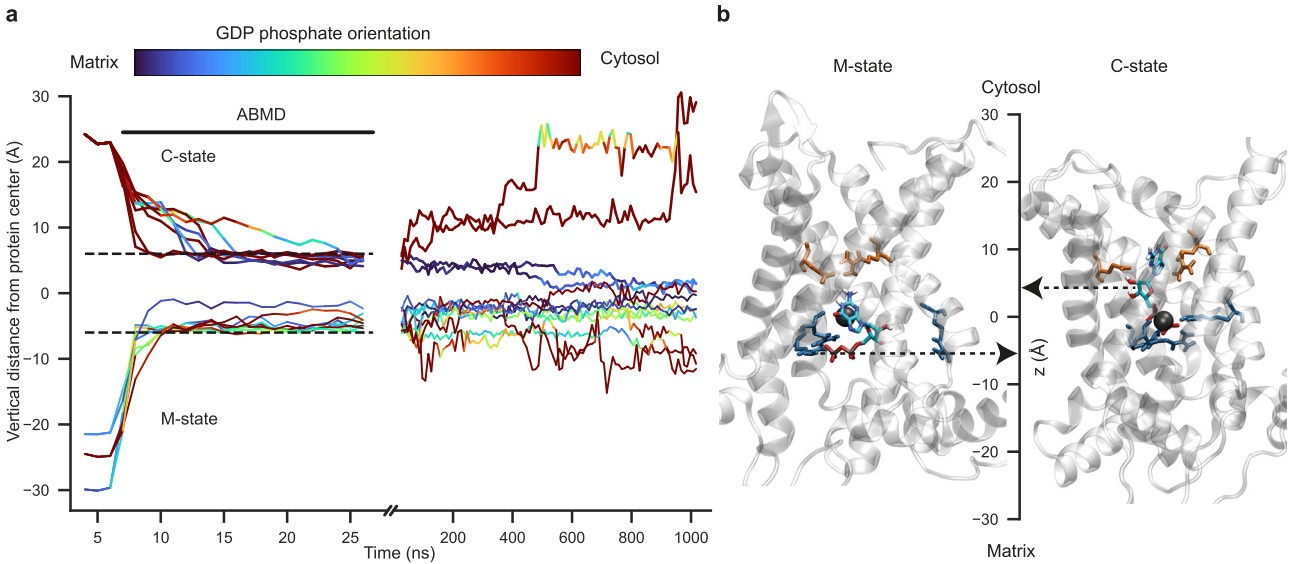

**Fig. 2 | Simulated binding pathways of GDP to UCP1 in the C-state and M-state conformations. a** Vertical distance of GDP center of mass from the protein center (in Å) as a function of simulation time. Colors indicate GDP orientation, computed as the cosine of the angle between the phosphate-base vector of GDP and the $z$-axis. Binding was accelerated using the Adiabatic Bias Molecular Dynamics (ABMD) method for 20 ns followed by 1 μs of unbiased relaxation. **b** Snapshots of UCP1 M-state (left) and C-state (right) with bound GDP. The center axis shows the vertical distance of GDP from the protein center (represented by a black sphere). UCP1 residues represented as sticks are R92-E191 (orange) and triplet RRR84 (blue). Source data are provided as a Source Data file.

Since AAC sequences share only 20% identity with rat UCP1, the homology models need to be carefully validated using external information. A model built using the automated machine-learned engine AlphaFold2 was analyzed for comparison. In addition, an independent homology model was built based on the recent cryo-EM structure of a putative mitochondrial carrier within mitochondrial complex IV of *Tetrahymena thermophila*[24]. This structure is a unique example of a putative mitochondrial carrier in a CATR-free native C-state conformation, with a sequence identity to UCP1 comparable to AAC.

A global comparison of these models indicates variations in the splay of helices on the cytoplasmic side (Fig. 1c), which tends to decrease upon physical relaxation in an all-atom MD simulation (Fig. 1d), converging towards an intermediate configuration between UCP1-7W5Z and UCP1-AAC. We have extended our investigation of the 7W5Z-based model by MD simulations, to 1-μs trajectories of apo UCP1. Analysis of these trajectories confirms that these models are close in structure and behavior to our C-state UCP1-AAC models and that they further converge during simulations. In particular, the structure of residues interacting with nucleotides in the common substrate binding sites is very similar between the two sets of homology models (RMSD below 2 Å, see Supplementary Fig. 1). Our conclusion is that both of those templates yield equivalent models for the purposes of this study. However, the carrier found in 7W5Z exhibits sequence divergence at the cytoplasmic salt-bridge network (Fig. 1). Next, we characterized the models by estimating their permeability to water in simulations (Supplementary Fig. 2). The NMR structure of UCP2 is known to be fully open and water-permeable[17], and the related UCP1 model (UCP1-NMR) follows the same trend, which is expected to be incompatible with controlled proton transport. In comparison, AAC structures and UCP1 models based thereon or on 7W5Z exhibit little or no water permeability. Finally, the QMEAN6[25] structure quality score was computed on the models at different stages of the simulation and compared to the experimental structures of AAC as reference (Supplementary Fig. 3). The C-state and M-state UCP1-AAC models have scores between those of the corresponding AAC crystal structures, indicating that the homology modeling did not significantly deteriorate the overall quality of the structures.

## GDP preferentially binds C-state UCP1, in an upright orientation

Multiple trajectories of GDP binding were obtained from Adiabatic Bias Molecular Dynamics (ABMD) simulations. This method enables minimally biased, rapid sampling of GDP binding by selecting thermodynamic fluctuations of the nucleotide that favor binding. To avoid making assumptions about nucleotide access through the cytosol or the matrix, we explored binding to the cavity in both the C-state and the M-state models. GDP was initially placed at the boundary between the intermembrane or matrix side and the UCP1 C-state and M-state models, respectively. The timelines of these trajectories are represented in Fig. 2a. GDP entry in the M-state conformation is rapid and direct, with no preferential orientation of GDP and no obstacle until it reaches the center of the protein and forms contacts with triplet RRR84 (Fig. 2b, left). In contrast, binding trajectories to the C-state conformation are more constrained and fall into well-separated categories. First, GDP tends to align along the transmembrane axis, leading to two orientation modes with the phosphate group pointing either toward the protein center or towards the cytoplasm opening (line colors in Fig. 2a). Second, some binding trajectories slow down around 11 Å from the protein center due to the presence of a salt bridge composed of R92 and E191 extending across the cavity and linking helix 2 with helix 4. Of note, this salt bridge was not present in any of the initial C-state models (homology or AlphaFold) but formed on the 100-ns timescale during the initial relaxation simulation, as helix splay on the cytoplasmic side relaxes (Fig. 1d). Supplementary Movie 1 shows that GDP can either disrupt this salt bridge or bypass it (an entropic bottleneck) to reach the center. A subset of these trajectories was extended with 1 μs unbiased relaxation simulations to allow GDP to explore the cavity and to assess the final pose stability at each endpoint. For the C-state conformation, two trajectories ending in each orientation of the phosphate group were selected for further relaxation. Only one of these orientations, with the diphosphate group facing the arginine triplet, is stable on the microsecond timescale, the phosphate group of GDP interacting with triplet RRR84 and the base moieties with residues F88 or W281 as well as R92 and E191 (Fig. 2b, right). In trajectories where GDP binds in the opposite orientation, spontaneous unbinding is observed within 1 μs. In the M-state conformation, 1 μs unbiased relaxation trajectories confirm the absence of

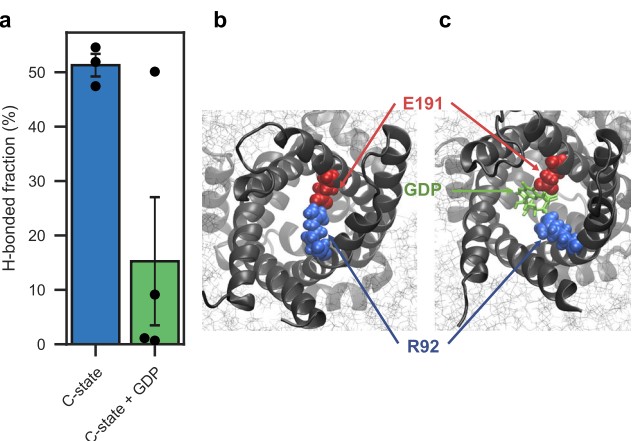

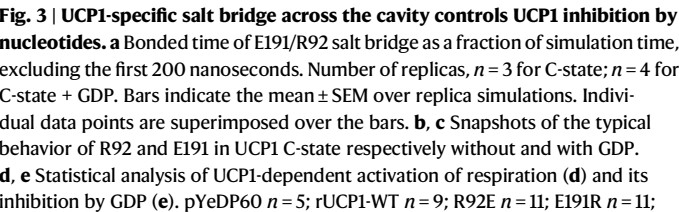

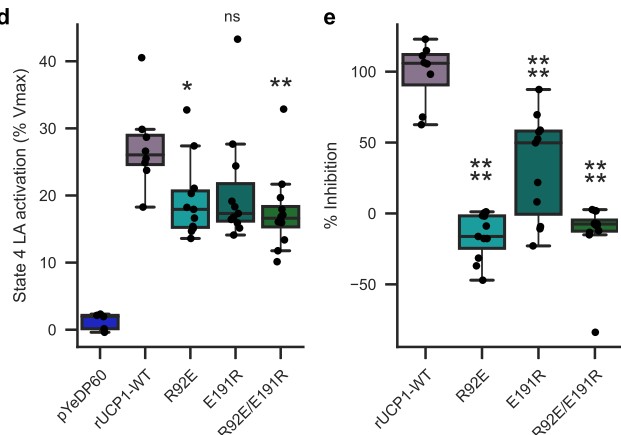

**Fig. 3 | UCP1-specific salt bridge across the cavity controls UCP1 inhibition by nucleotides. a** Bonded time of E191/R92 salt bridge as a fraction of simulation time, excluding the first 200 nanoseconds. Number of replicas, $n = 3$ for C-state; $n = 4$ for C-state + GDP. Bars indicate the mean ± SEM over replica simulations. Individual data points are superimposed over the bars. **b, c** Snapshots of the typical behavior of R92 and E191 in UCP1 C-state respectively without and with GDP. **d, e** Statistical analysis of UCP1-dependent activation of respiration (**d**) and its inhibition by GDP (**e**). pYeDP60 $n = 5$; rUCP1-WT $n = 9$; R92E $n = 11$; E191R $n = 11$;

R92E/E191R $n = 11$ over three biologically independent experiments. LA = Lauric acid. Boxes of the box plots indicate the first quartile, the median, and the third quartile. The whisker length is 1.5 times the interquartile range. Individual data points are superimposed over each boxplot. Data were analyzed by one-way ANOVA using Dunnett's post-test. ns = not significant, *$p$-value ≤ 0.05, **$p$ ≤ 0.01, ***$p$-value ≤ 0.001, ****$p$-value ≤ 0.0001. Statistical analyses are presented in Supplementary Tables 3 and 4. Source data are provided as a Source Data file.

preferential orientation of the base moiety within the cavity. In further extended simulations, GDP unbinds on the microsecond timescale in two out of three simulations (Supplementary Fig. 4). Therefore, further analysis of GDP-UCP1 interactions was performed on the C-state only.

### Molecular determinants controlling UCP1 inhibition

To obtain a more precise map of interactions of bound GDP with UCP1 C-state and account for possible effects of GDP binding on UCP1 conformation, simulations were run with the initial model bound to GDP from the onset in poses based on clustering analysis of the binding simulations. Respectively, three and four simulations were run for 2 μs for the *apo* and GDP-bound models.

**Identification of a conserved pair of charged residues involved in GDP inhibition.** Simulations of the *apo* C-state UCP1 model show the formation of the salt bridge between R92 and E191, which partially obstructs the cavity on the cytosolic side (Fig. 3a, b), hence its effect on GDP binding trajectories. However, in the presence of GDP in the cavity, the R92-E191 salt bridge is significantly reduced (Fig. 3a). Instead, strong interactions between the R92 and E191 residues and the nucleotide are observed (Fig. 3c). To experimentally address the function of these interactions, UCP1 wild-type (WT) and UCP1 mutants were expressed in yeast mitochondria, and UCP1-mediated uncoupling of respiration was assayed. The following mutations were introduced in rat UCP1: R92E or E191R to prevent the salt bridge formation, and the charge-swap double mutation R92E/E191R to attempt the salt bridge restoration by charge inversion. Expression levels of all mutants were assessed by immunological detection of recombinant rUCP1 and endogenous VDAC (see Supplementary Fig. 5 and Supplementary Table 1). Respiratory coupling ratios (RCR) of yeasts expressing rUCP1-WT or mutants displayed no significant difference (Supplementary Table 2).

Spheroplasts expressing UCP1-R92E or UCP1-R92E/E191R show a significant decrease of lauric acid (LA)-induced activation of respiration to 45% $VO_{2max}$ while UCP1-E191R behaves as UCP1-WT (Fig. 3d). In spheroplasts expressing UCP1-R92E and UCP1-R92E/E191R, GDP does not inhibit LA-induced respiration increase. In contrast, for the UCP1-E191R mutant, GDP partially inhibits (21%) LA-induced respiration (Fig. 3e, see Supplementary Tables 3 and 4 for statistical analysis).

Accordingly, single mutations of the salt bridge demonstrate the function of both amino acids in controlling UCP1 inhibition by nucleotides. Unexpectedly enough, charge inversion of the salt bridge does not restore the wild-type phenotype. However, this is explained by simulations of the UCP1-R92E/E191R model showing collective salt bridge reorganization. Consequently, E92 establishes a stable salt bridge with arginine residues of triplet RRR84 at the expense of the E92-R191 salt bridge (Supplementary Fig. 6).

**UCP1-nucleotide contacts obtained from simulations.** To get a comprehensive view of how GDP interacts with UCP1 within the common substrate binding site of mitochondrial carriers, we performed a statistical analysis of intermolecular contacts (Fig. 4a). In addition, clustering was performed on GDP positions of MD simulations. The representative configuration of the main cluster (54% of the configurations) is shown in Fig. 4b. The nucleotide is anchored on two sides, with polar and non-polar interactions with the base and ribose moieties and five positively charged residues forming permanent or transient salt bridges with the phosphate group.

R92 and E191 have a high contact probability with GDP, E191 forms specific H bonds with the base of GDP, and R92 binds the phosphate group, the ribose, and the base alternatively. Residues of triplet FIW88 are engaged in frequent contact with the base or the ribose of GDP, especially W281, which forms aromatic interactions ("pi stacking") with the purine base. The list of contacts also includes two residues from asymmetric triplet QNL85[10,13], Q85, and L278. Arginine residues from triplet RRR84 have the highest contact probability, and the phosphate group remains close to the triplet during the entire duration of the three replicas, indicating a strong interaction. Lysine residues from the salt bridge network on the matrix side (K38 and K138) are also involved in electrostatic interactions with the phosphate group.

Of note, interactions between GDP and phospholipids were observed in MD simulations. The phospholipids enter the cavity through the gaps between transmembrane helices and form hydrogen bonds with the base. Phospholipids bind the same GDP atoms as E191, which lowers the binding rate of this residue. This mainly occurs in one of the replicas, where a POPE molecule binds GDP during the whole simulation: this increases the observed variance of contact times without affecting the list of contact residues. Of note, we did not

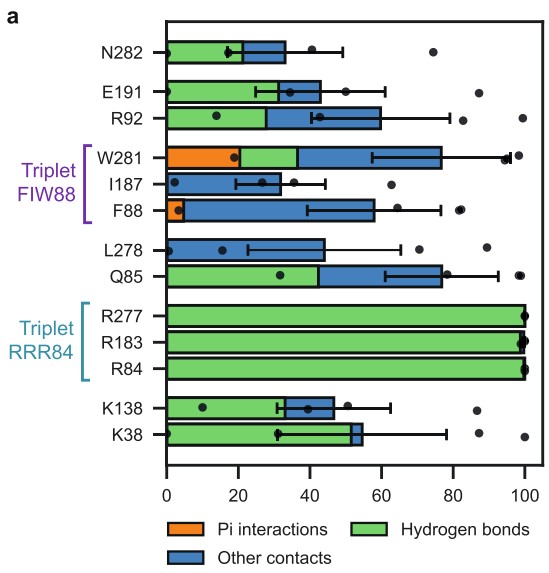

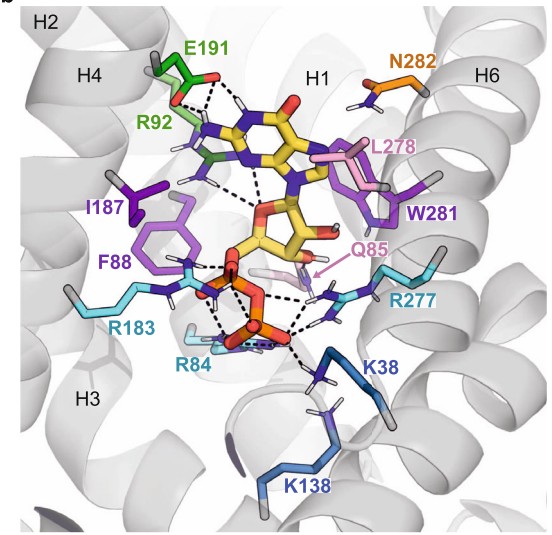

**Fig. 4 | Main interactions between GDP and UCP1 residues in simulations.**
**a** Contact time between GDP and protein residues as a fraction of simulation time excluding the first 200 ns. The part of that contact time involving hydrogen bonds is represented in green, and the part involving pi interactions is represented in orange. Total contact times and their components are averaged over the trajectories. The number of replicas is $n = 4$. Bars indicate the mean ± SEM of total contact

times. The individual contact time of each replica is represented by a black dot. **b** Snapshot of the representative configuration of the main GDP cluster (54% of the configurations, one out of three clusters). GDP is depicted as thick sticks with carbons in yellow. Protein residues are shown as thin sticks and are colored by triplets. Hydrogen bonds are represented as black dashed lines. Source data are provided as a Source Data file.

observe such intrusions of the phospholipids inside UCP1 cavity in the *apo* simulations.

**Triplet FIW88 is critical for the control of UCP1 activity.** Since triplet FIW88 also is in contact with GDP in simulations and is located between the R92/E191 pair and triplet RRR84, one helix turn away from both, we investigated its role in GDP inhibition of UCP1 activity by site-directed mutagenesis. The following apolar mutations were introduced in rat UCP1: FIW88AAA, F88A, I187A, and W281A. Expression levels of all mutants were assessed by immunodetection of both recombinant UCP1 and endogenous mitochondrial VDAC (Supplementary Fig. 5 and Supplementary Table 1).

The superposition of the five curves shows three distinct phenotypes (Fig. 5a). Spheroplasts expressing rUCP1-FIW88AAA exhibit an increase in respiration in response to LA and a complete loss of respiration inhibition by GDP. Decomposition of the triplet mutant in single mutations reveals that rUCP1-F88A has a respiration profile identical to rUCP1-WT. In contrast, rUCP1-I187A and rUCP1-W281A have increased respiration in response to LA and 75 % loss of respiration inhibition by GDP (Fig. 5c and Supplementary Tables 5 and 6 for statistical analysis). Simulations of the rUCP1-FIW88AAA mutant with bound GDP show a displacement of the base moiety of GDP and an increase in hydration of the site by about 8 water molecules compared with the wild-type. In particular, increased hydration points to a possible indirect role of I187 in GDP binding, whereby it limits the amount of water in the cavity, which would otherwise compete with GDP for polar interactions with the protein.

To illustrate the lack of regulation of the rUCP1-FIW88AAA mutant in a closer cellular context, GDP was added before LA. Figure 6 shows that, after inhibition of rUCP1-WT with GDP, the addition of LA hardly stimulates UCP1-mediated respiration uncoupling. In contrast, rUCP1-FIW88AAA rapidly increases respiration uncoupling in response to LA, even in the presence of GDP.

**UCP1 inhibition is insensitive to FA chain length and to the number of nucleotide phosphates.** Long-chain fatty acids have been shown to better activate UCP1 that medium-chain FA[26]. In order to rule out

possible effects of FA chain length on our results, we performed respiration experiments with palmitic acid (C16), a widely used activator of UCP1. We find slightly higher activation by palmitic acid (Supplementary Fig. 7d), as described in[26]. Despite this increase in activation, rUCP1-WT or variants show similar phenotypes of inhibition by GDP (Supplementary Fig. 7a, b).

To assess the importance of phosphates in UCP1-nucleotide interactions, we measured GTP inhibition of UCP1 after activation by lauric or palmitic acid. We find almost no significant difference between inhibition by GDP and GTP, after activation with either type of fatty acid, for either the wild-type or the variants documented above. In the case of rUCP1-WT, inhibition by GTP in response to palmitic acid as opposed to lauric acid is smaller by 20 percentage points. We also investigated UCP1-GTP interactions by MD simulations (Supplementary Fig. 7c). GTP and GDP exhibit similar contacts with UCP1, especially between di/triphosphate moiety and triplet RRR84 at the bottom of the cavity. Contact times with key residues identified in the GDP simulation were mainly unchanged, showing the specificity of interactions in this region; the main difference is a reduced interaction time with Q85.

**Molecular bases for purine nucleotide specificity of UCP1 inhibition.** It has been shown that pyrimidine nucleotides do not inhibit UCP1-mediated uncoupling of respiration[27]. In simulations, UDP and GDP show similar contacts between the phosphate and ribose moieties and residues of the UCP1 cavity but not with the base moiety of UDP (Fig. 7). UDP does not bind E191, W281, or Q85 but preferentially binds N184 and N188, which have little interaction with purine nucleotides. However, GTP rarely binds Q85 (Supplementary Fig. 7), thus E191 and W281 seem to be the main residues responsible for the nucleotide selectivity.

## Discussion

Here we have used UCP1 simulations and functional assays to explore how nucleotides inhibit UCP1 uncoupling activity. The convergence of structural models from different origins under simulation indicates that AAC-based models of UCP1 C-state are biased towards excessive

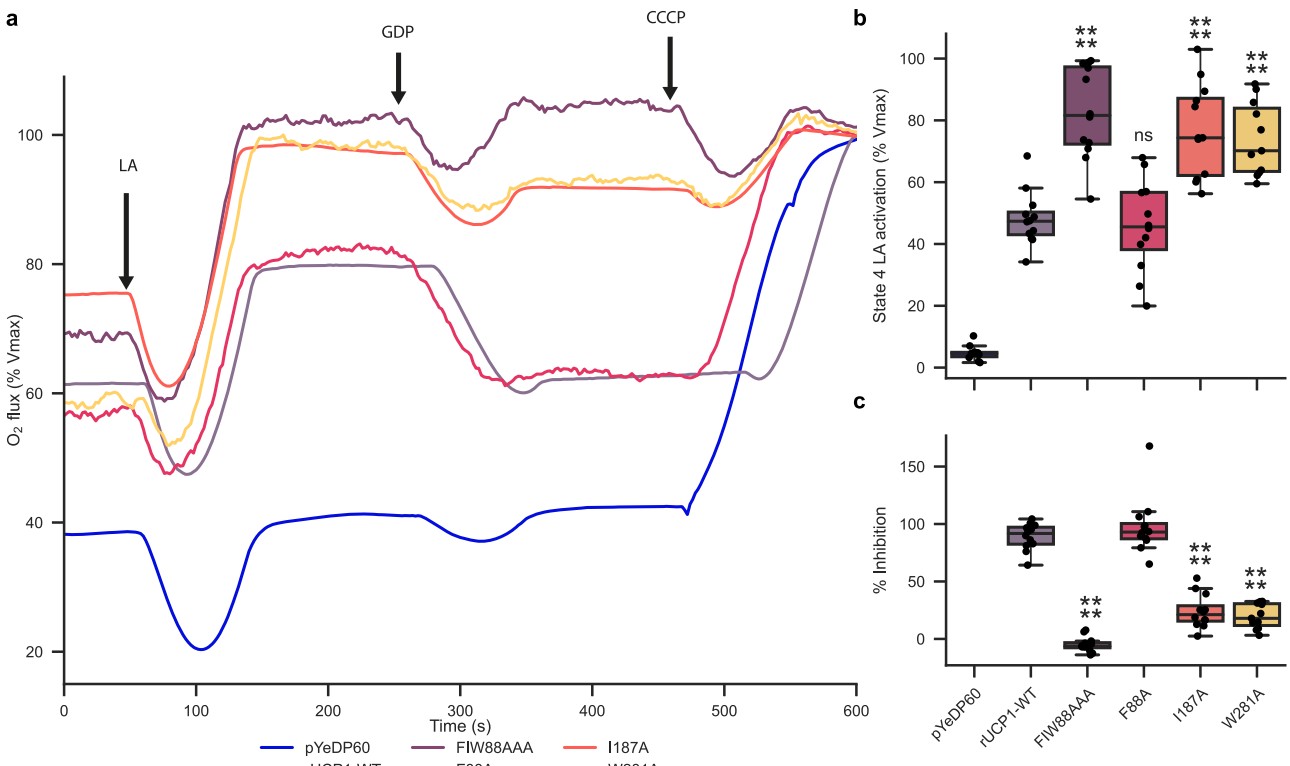

**Fig. 5 | Aromatic residues within the common substrate binding site control UCP1 regulation. a** Representative oxygen flux consumption curves of permeabilized spheroplasts harboring either plasmid expression rUCP1-WT or mutants. **b** UCP1 activation in response to LA. **c** Inhibition by GDP of UCP1-dependent increase of respiration after adding LA. pYeDP60 $n = 10$; rUCP1-WT $n = 11$; FIW88AAA $n = 11$; F88A $n = 11$; I187A $n = 11$; W281A $n = 11$ over three biologically independent experiments. LA = Lauric acid. Boxes of the boxplots indicates the first quartile, the median and the third quartile. The whisker length is 1.5 times the interquartile range. Individual data points are superimposed over each boxplot. ns = not significant, * = $p$-value ≤ 0.05, ** = $p ≤ 0.01$, *** = $p$-value ≤ 0.001, **** = $p$-value ≤ 0.0001. Data were analyzed by One-way ANOVA using Dunnett's post-test. Statistical analyses are presented in Supplementary Tables 5 and 6. Source data are provided as a Source Data file.

opening on the C side and that physics-based relaxation of UCP1 models tends to recover the native shape.

The SLC25 family of alpha-helical mitochondrial transporters is challenging to study in vitro because they are highly prone to inactivation by detergent[17,28]. Therefore, we have chosen to assess predictions from molecular modeling directly by respiration assays of recombinant yeast spheroplasts to avoid the pitfall of inactivating UCP1 protein in detergent solution. Both computational and biochemical approaches converge toward a mechanism for UCP1 inhibition by GDP, in which interactions with residues R92 and E191 are critical.

R92 and E191 residues have been previously studied in the context of the pH dependence of UCP1 nucleotide binding. The R92T mutation lowers GTP affinity and completely abolishes the pH sensitivity of nucleotide binding to UCP1[29] whereas E191Q only partially modifies this pH sensitivity, resulting in an increased GTP affinity at pH higher than 6.8[30]. Martin Klingenberg postulated the existence of a salt bridge involving E191 acting as a pH-dependent gate for nucleotide access[31]. Our numerical study confirms both the existence of this salt bridge and its interference with GDP binding. Analysis of simulated trajectories of GDP binding shows either disruption or bypass of the R92-E191 salt bridge to enter the cavity. Once GDP relaxes inside the binding site, the salt bridge dissociates, allowing R92 and E191 to interact directly with the nucleotide. Although charge inversion in the E92-R192 mutant did not restore the wild-type phenotype, MD simulations of this mutant show that charge inversion results in a reorganized network of salt bridges around R191, explaining the lack of phenotype restoration. Sequence alignment of mitochondrial carriers (Supplementary Fig. 8) shows that the R92-E191 salt bridge is almost uniquely restricted to the

subfamily of UCPs (UCP1, UCP2, UCP3, UCP4, and UCP5). Only one other human mitochondrial carrier, SCMC1, which transports ATP-Mg$^{2+}$ against phosphate, contains charged residues equivalent to R92/E191.

One helix turn below triplet R92, residues of the cavity-facing triplet FIW88, primarily F88 and W281, establish contacts with the base and ribose moiety of the nucleotide (Fig. 4). W281, in particular, gives a strong signal both in experiments and simulations. The triplet mutation FIW88AAA in UCP1 not only suppresses GDP inhibition of respiration uncoupling but also exhibits an increased FFA-dependent respiration uncoupling activity, insensitive to the initial addition of GDP. Molecular dynamics simulations of AAC showed the binding of the fatty acid via a fenestration between transmembrane helices 5 and 6, leading to water molecules leaking across the membrane[32]. That work identified a specific contact between FFAs and AAC residue Y186. Of note, residues I187 and E191 of UCP1 are replaced by highly conserved Y186 and Y190 residues in AAC, which have been described as part of a tyrosine ladder by Pebay-Peyroula and colleagues[6].

At the end of the cavity, one helix turn away from triplet FIW88, the phosphate moiety of the nucleotide forms a stable interaction with triplet RRR84, which corresponds to triplet RGR88 of AAC also documented as a nucleotide anchor point[33,34]. Similar interactions have been observed in simulations of ADP binding to AAC[8,9] and UCP2[22].

Our results complete previous knowledge to highlight differences between AAC and UCP1 with respect to nucleotide binding. Previous mutagenesis results on yeast AAC have highlighted several positively charged residues in the C-state cavity[35,36], which are essential for AAC transport function and replaced by uncharged residues in the UCP1 sequence (Fig. 1). In addition, UCP1 also lacks asymmetric triplets

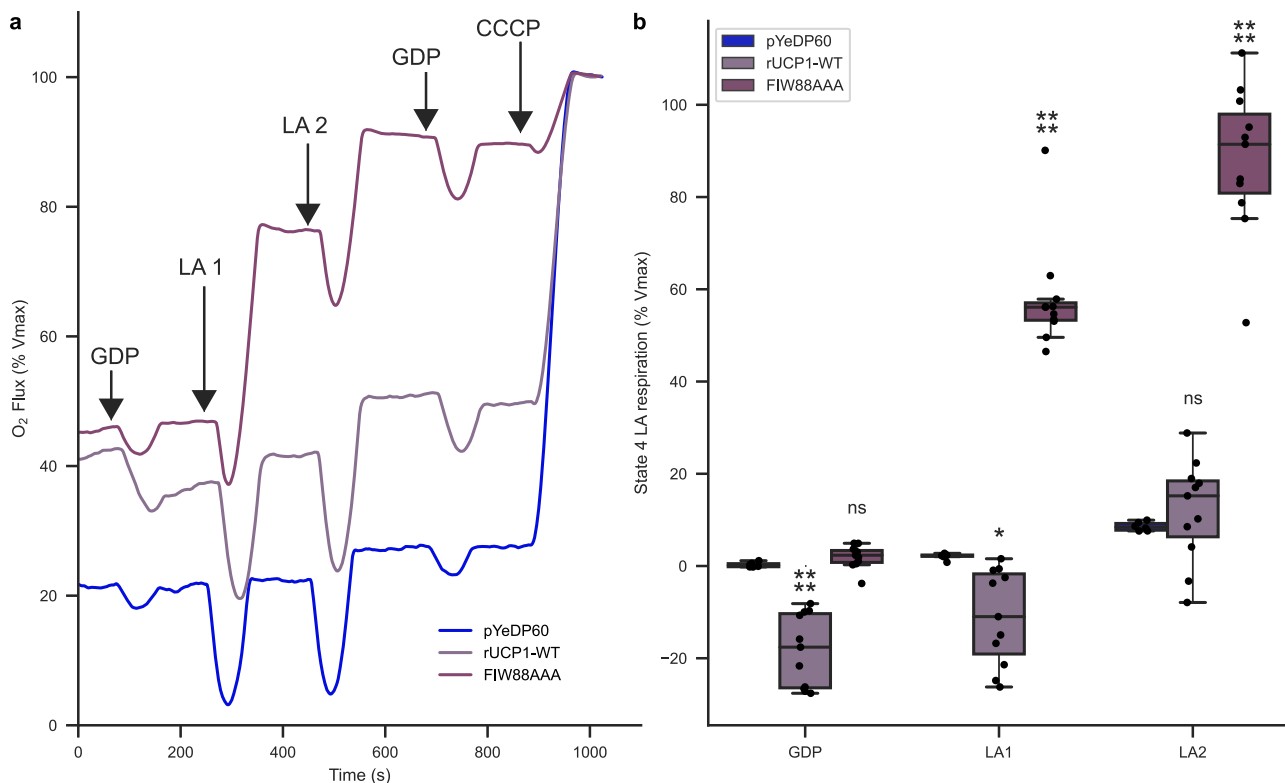

**Fig. 6 | rUCP1-FIW88AAA responds to LA and uncouples respiration in the presence of GDP. a** Experimental curves of permeabilized spheroplasts harboring either rUCP1-WT or rUCP1-FIW88AAA. Arrows represent the different injections. **b** Effect of GDP and of two lauric acid additions after GDP inhibition. pYeDP60 $n = 6$; rUCP1-WT $n = 11$; FIW88AAA $n = 11$ over three biologically independent experiments. LA = Lauric acid. Boxes of the box plots indicates the first quartile, the median and the third quartile. The whisker length is 1.5 times the interquartile range. Individual data points are superimposed over each boxplot. Data were analyzed by one-way ANOVA using Dunnett's post-test. ns = not significant, * = $p$-value ≤ 0.05, ** = $p$ ≤ 0.01, *** = $p$-value ≤ 0.001, **** = $p$-value ≤ 0.0001. Statistical analysis is presented in Supplementary Table 7. Source data are provided as a Source Data file.

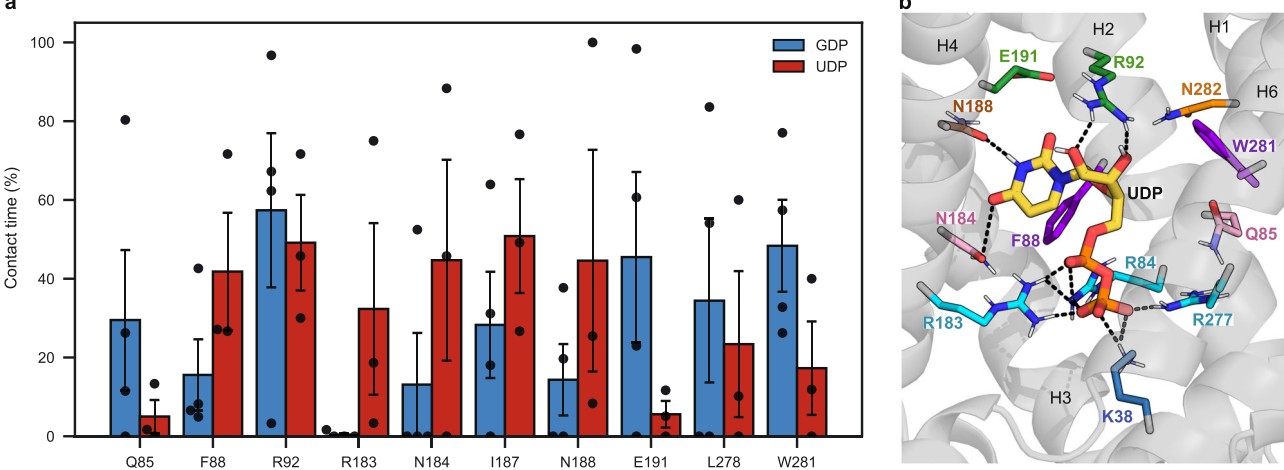

**Fig. 7 | UDP exhibits reduced binding to Q85, E191, and W281 in simulations. a** Contact times between nucleotide bases and UCP1 residues. Only the residues with the highest contact times are represented. Contact times are measured from 200 ns to 800 ns and averaged over the trajectories. Number of replicas, $n = 3$ for UDP and $n = 4$ for GDP. Bars indicated the mean ± SEM between replicas. Individual contact time of each replica is represented by a black dot. **b** Snapshot of the representative configuration of the main UDP cluster (40% of the configurations, one out of three clusters). GDP is depicted as thick sticks with carbons in yellow. Protein residues are shown as thin sticks. Hydrogen bonds are represented as black dashed lines. Source data are provided as a Source Data file.

that are critical for substrate selectivity within the common substrate binding site[13]. A critical example is the asymmetric triplet RGR83 in AAC, whose UCP counterpart is the symmetric triplet RRR84. All three arginines are known to be essential for UCP1 inhibition[29,37,38]: indeed, we find that the phosphate moiety of GDP or GTP consistently binds all three arginine residues (Fig. 4). In contrast, in AAC, asymmetric placement of the two arginines determines an asymmetric position of the nucleotide[8,9,34]. In a recent conceptual model of nucleotide translocation by AAC, the small size of G199 leaves a gap that can accommodate the base moiety (Figure 2a in ref. 5). This would allow the phosphate

groups to flip to the matrix side, which is expected to be necessary for transport[36]. Substitution of this glycine with R183 in UCP1 closes this gap and results in a much tighter ring of residues. We also find that additional interactions (mainly R92, E191, I187, and W281) stabilize GDP in an orientation orthogonal to the plane of triplet RRR84, further preventing any movement of the base towards the matrix side. Altogether, these differences provide a basis for explaining the lack of nucleotide transport by UCP1 as opposed to AAC.

Stimulating energy expenditure through respiratory uncoupling has been proposed in a therapeutic context for metabolic diseases[2,3]. Search for UCP1-activating drugs other than FFAs has been attempted[39]. The drug will need to cross several barriers to reach the inner mitochondrial membrane and overcome the natural inhibition of UCP1 by purine nucleotides. An alternative approach is to design unregulated variants of UCP1 to minimize the need for activating drugs or FFAs. In contrast to E191R and R92E, alanine variants of triplet FIW88 fulfill those criteria: they entirely escape nucleotide inhibition and display increased basal and FFAs-stimulated uncoupling activity in the cellular context of high nucleotide concentration. They may therefore represent a promising variant for adipocyte browning to stimulate energy metabolism in engineered human adipocytes.

## Methods

### Modeling and simulations

**Structural models.** Homology models of rat UCP1 were based on two high-resolution X-ray crystal structures of AAC, one in a C-state conformation (*Bos taurus*, PDB entry 2C3E[23]) and the other in the M-state conformation (*Thermothelomyces thermophilus*, PDB entry 6GCI[11]). Sequences were aligned using MAFFT[40] and rendered using Jalview[41]. Two homology models were created using Modeller[42] from the two AAC conformations. Some restraints on secondary structures were added in Modeller at the termini of the protein (residues 7-14 and 296-301) to keep these parts of the protein in an α-helix configuration. Three cardiolipin molecules were placed around the models according to their positions in the crystal structures of *Bos taurus* AAC. Three cardiolipin molecules partially modeled in crystal structures were kept and embedded in membranes with homology models (five cardiolipin molecules are partially resolved in the AAC M-state crystal structure but only three were kept in the initial model for consistency with the C-state model). The UCP1 C-state model created from the alignment of Fig. 1 showed instabilities in helix 6 during relaxation simulations, resulting in the loss of secondary structure around L278. L278 is an insertion in UCP sequences regarding AAC sequences, and Modeller could not create a stable α-helix AAC C-state with a gap on L278. Manually removing the gap in the AAC sequence highly improves the stability of helix 6 in UCP1 C-state simulations. This results in a rotation of the end of the helix by 1/4 of the turn, yet this rotation relaxes during simulations without disrupting the helix. In contrast, M-state models were built using the original alignment because it yielded a stable helix 6 in simulations, whereas the helix rotation created when the gap was removed disrupted the salt bridge network on the cytosolic side, a key feature for the stability of the M-state.

As a control, a homology model of rat UCP1 based on the NMR structure of *Mus musculus* UCP2[16] (PDB entry 2LCK) was produced. The whole UCP1 sequence was used to build the model and three cardiolipin molecules were manually placed around the model at a similar position to that in the UCP1 C-state model.

The sequence of the putative mitochondrial carrier from the Cryo-EM structure 7W5Z chain M2 (Uniprot entry Q23M99) was aligned on rat UCP1 with MAFFT. After manual adjustments to the alignment, Modeller was used to generate the UCP1-7W5Z model. The UCP1-AlphaFold model was retrieved from the AlphaFold Protein Structure Database (UniProt P04633).

## Molecular dynamics simulations

**Simulation setup.** Membrane-embedded, solvated simulation models of UCP1 *apo* M-state and C-state were built using CHARMM-GUI membrane builder[43]. $100 \times 100$ Å$^2$ membranes were built with the following composition: on the cytosolic side the leaf composition is 50% POPC, 40% POPE, and 10% CL, on the matrix side the leaf composition is 40% POPC, 30% POPE and 30% CL. The systems were assembled with a salinity of 150 mM of KCl and hydrated by around 14400 TIP3P[44] water molecules. Proteins were prepared with protonated lysine and arginine, neutral histidine and cysteine, and deprotonated glutamate and aspartate residues. CHARMM-GUI standard minimization and equilibration were used for the M-state model (total restrained equilibration time of 1.85 ns). Due to secondary structure instabilities on the 2nd helix of the UCP1 C-State model, the restrained equilibration time was extended to a total of 30.75 ns. After equilibration, the size of the simulation box is around $98 \times 98 \times 84$ Å$^3$ for C-state simulations and around $99 \times 99 \times 87$ Å$^3$ for M-state simulations. GDP was used as an inhibitor to match experimental conditions. Whenever GDP was included, it was modeled in the GDP$^{3-}$ protonation state.

**Molecular dynamics simulations.** All MD simulations were performed using the NAMD software[45] with the CHARMM36m force field[46] in the NPT ensemble at 310 K, 1.013 bar, and periodic boundary conditions. Underdamped Langevin dynamics with a coupling coefficient of 0.1 ps$^{-1}$ was used to control temperature, without coupling for hydrogen atoms. The pressure was maintained using the Langevin piston algorithm with an oscillation period of 200 fs and a damping timescale of 100 fs.

All simulations used hydrogen mass repartitioning (HMR)[47] and a timestep of 4 fs. Three independent simulations of 2 μs were performed for UCP1 C-state *apo* and three independent simulations of 1 μs were performed for UCP1 M-state *apo*. Simulation trajectories were visualized with VMD[48].

Equilibration, and simulation of the UCP1-NMR, UCP1-7W5Z, and UCP1-AlphaFold models were performed using the same protocol as UCP1 M-state simulations. A single-replica, 1 μs simulation was performed for UCP1-NMR, three replicas, 1 μs simulations for UCP1-7W5Z, and a single-replica, 300ns simulation for UCP1-AlphaFold.

AAC simulations were started from the X-ray structures (2C3E for the C-state and 6GCI for the M-state). Residues from G253 to V256, which are not resolved in the 6GCI structure, were reconstructed with Modeller. System preparation and simulation protocols were identical to those of UCP1 M-state simulations. One replica of each model was simulated for 1 μs.

**Accelerated binding simulations.** To explore possible GDP binding trajectories and poses, simulations were run wherein binding was accelerated by non-equilibrium biases following the Adiabatic Bias Molecular Dynamics (ABMD) method[49]. The Tcl-scripted implementation of ABMD in the Colvars Module[50] was used and can be found here: https://github.com/Colvars/colvars/blob/master/colvartools/abmd.tcl.

To prepare the starting points of ABMD simulations of GDP binding, GDP$^{3-}$ was placed at the entrance of the UCP1 cavity in one position on the cytoplasmic side for C-state simulations and in three different positions on the matrix side for the M-state simulations.

ABMD was applied to the distance between the nucleotide and the protein center, with force constant 10 kcal/mol/Å$^2$ and a final restraint level of 6 Å from the protein center. The protein center was defined as the center of a set of 18 alpha carbons, of the A27 to I29 triplets (on odd-numbered helices) and the Q83 to Q85 triplets (on even-numbered helices). This included the arginine triplet RRR84. After 20 ns under ABMD bias, simulations were continued without any bias for a total of 1 μs to allow for free exploration of the cavity by GDP. Nine replicas were carried out for C-state and four for each starting position

for M-state simulations. ABMD simulations started from an early iteration of the models, which included full-length N and C-termini and the initial alignment of helix 6 (see Homology Modeling above). These differences are not expected to affect the specific results on GDP binding to the central cavity of UCP1.

**Equilibration with nucleotide.** Nucleotide molecules were embedded inside UCP1 C-state and M-state homology models before equilibration, according to the estimated positions from the clustering on ABMD simulations. Simulations were performed with the same protocol as the *apo* model simulations. Respectively, four and three independent simulations of 2 μs were performed for UCP1 C-state + GDP and UCP1 M-state + GDP. Three independent simulations of 0.8 μs were performed for UCP1 C-state + UDP.

**Simulations with GTP.** A GTP molecule was placed near the entry of the protein in the UCP1 *apo* systems at 1 μs of simulation (potassium ions were added randomly in water to negate the charge). Ten ABMD simulations were performed for each replica to accelerate nucleotide entry into the protein with the same parameters as those used for GDP. For each of the three replicas, one ABMD trajectory where the phosphate groups reach the R84 triplet was selected as a starting point of relaxation simulations. Then, the simulations were continued without bias for 1 μs.

**UCP1 FIW88AAA simulations.** The final protein structures of three UCP1 C-state trajectories with GDP were mutated in UCP1 FIW88AAA after 1 μs of relaxation. Then, simulations were continued during 1 μs.

### Analysis of simulation trajectories

**Charts and computations.** Charts were rendered using Matplotlib[51] and Seaborn[52]. Boxes of the boxplots indicates the first quartile, the median, and the third quartile. The whisker length is 1.5 times the interquartile range. Computations were performed with MDAnalysis[53,54] and VMD[48].

**Water permeability.** Water permeability was estimated by counting the number of water molecules which went through the membrane inside or near the protein. To detect such permeation events, the membrane was split into three regions of equal thickness along the $z$ axis. A permeation event was detected if a water molecule went from one side of the membrane to the other, traveling through the three regions of the membrane and staying within 15 Å of the protein. This criterion eliminates false positives due to water molecules diffusing across periodic boundaries.

**Helix splay profiles.** The average alpha-carbon positions $\mathbf{r}_i$ of the transmembrane helices in the membrane plane $(X, Y)$ were computed a running 12 Å-wide window. From these average positions, the helix splay profile was computed as an in-plane radius of gyration:

$$R_{XY}(z) = \sqrt{\frac{1}{N}\sum_{i}^{N}|\mathbf{r}_i(z) - \bar{\mathbf{r}}(z)|^2} \qquad (1)$$

where $z$ is the position of the window, $N$ the number of transmembrane helices and $\bar{\mathbf{r}}$ the average of $\mathbf{r}_i$ over all helices.

**Contact detection.** A contact between two groups of atoms was detected if at least one pairwise distance between the atoms of the two groups is lower than 3 Å. A hydrogen bond was detected if the distance between donor and acceptor atoms was less than 3.3 Å and the angle donor/hydrogen/acceptor was higher than 140°. A pi interaction was detected if at least three atoms of an aromatic ring of a protein residue were closer than 3.8 Å to the aromatic rings of the nucleotide. A salt

bridge was defined as bonded if the two residues formed at least one hydrogen bond.

**Clustering.** Clustering of GDP positions was performed with either TTClust[55] or the scikit-learn library[56]. Nucleotide positions were clustered after alignment of the transmembrane helices using an agglomerative clustering followed by a K-Means clustering.

### In vivo respiration assays
All reagents were purchased from Sigma-Aldrich unless specified in the text.

**Mutagenesis and yeast transformation.** The pYeDP60 yeast expression vector harboring rat UCP1 gene with an N-terminal (*His*)$_8$-tag and a TEV cleavage site was previously validated for the functional expression of UCP1 in yeast (W303.1b) mitochondria[18]. Mutagenesis by gene synthesis was performed by Twist Bioscience and UCP1 cDNA was subcloned in the pYeDP60 vector. Mutations in UCP1 were verified by DNA sequencing of the recombinant vector (Eurofins).

Yeast transformation was performed following the lithium acetate/single-stranded carrier DNA polyethylene glycol method[57]. In brief, Saccharomyces cerevisiae strain W303-GAL4 was grown overnight in YPDA medium (yeast extract 1%, adenine sulfate 540 μM, peptone 2% (BD Biosciences), glucose 2%) at 30 °C under 200 rpm agitation. Preculture (500 μL) was harvested by centrifugation at 5000 x *g* for 2 min and washed twice with TE buffer (Tris-HCl 10 mM pH 7.5, EDTA 1 mM). The yeast pellet was resuspended with the transformation mix (500 μL PEG4000 40% (Fluka), LiOAc 100 mM, DMSO 774 μM, YeastMaker DNA carrier 50 μg (Takara Bio), supplemented with 1 μg of pYeDP60 vector). The mixture was carefully homogenized by up and down pipetting 30 times and by shaking for 15 min at room temperature. The suspension was submitted to heat shock for 15 min at 42 °C. Cells were centrifuged for 2 min at 1000 x *g* and washed three times in TE buffer. Finally, pellets were resuspended with 90 μL of TE buffer and spread on an SDAA plate (yeast nitrogen base without amino acid 0.67%, tryptophan 195 μM, agar 2% (Sigma), and casamino acids 0.5% (USBio)). Plates were incubated for three days at 30 °C.

**UCP1 expression in yeast and spheroplasts preparation.** Yeast preculture was done overnight at 30 °C under 200 rpm agitation in S-lactate medium (lactate 2 %, yeast nitrogen base without amino acids 0.67%, 0.1% casamino acids, $(NH_4)_2SO_4$ 9 mM, $KH_2PO_4$ 7.3 mM, tryptophan 98 μM, pH 4.5 adjusted with NaOH) supplemented with 0.1% glucose. Preculture was diluted to $OD_{600nm} = 0.03$ in S-lactate medium final volume, supplemented with 0.1% glucose, and yeast cells were grown for 20 hours at 30 °C under 200 rpm agitation. Medium was exchanged with a new S-lactate medium, supplemented with 1% galactose to induce UCP1 expression, and cells were further grown for 4 h. Yeast cells were harvested by centrifugation (5000 x *g* for 10 min), washed in water, and resuspended (10 mL/g of wet yeast pellet) in SED buffer (sorbitol 1 M, EDTA 25 mM, and 1.4-dithiothreitol 50 mM). Cells were incubated for 10 minutes at room temperature, then washed with zymolyase buffer (sorbitol 1.2 M, $KH_2PO_4/K_2HPO_4$ 20 mM, pH 7.4), and resuspended (10 mL/g of yeast cells) in zymolyase buffer supplemented with 0.5 mg/mL Zymolyase-20T (Amsbio). After 45 min incubation at 30 °C, the resulting spheroplasts were washed three times (1000 x *g* centrifugation, 5 min) with respiration buffer (sorbitol 1 M, EDTA 0.5 mM, $MgSO_4$ 2 mM, NaCl 1.7 mM, $KH_2PO_4/K_2HPO_4$ 10 mM, Bovine Serum Albuvine fatty acids free 0.1%, pH 6.8). Finally, spheroplasts are stored at 4 °C in a respiration buffer (2 mL/g) for a night.

**Extraction of total yeast proteins and mitochondria preparation.** For mitochondria preparation, spheroplasts were broken using a Potter-Elvehjem grinder by moving up and down the pestle 15 times in TES buffer (10 mM Tris-HCl, 1 mM EDTA, 250 mM Sucrose, pH 7). After

centrifugation (800 x $g$ for 10 min at 4 °C), the supernatant was centrifuged for 30 min at 11,000 x $g$. The mitochondria pellet was resuspended with 100 µL of TES and quantified using the BCA method.

For total yeasts proteins extracts, yeasts were grown according to the previous paragraph in S-Lactate with 4 h of induction with 1% galactose. A volume of culture corresponding to an $OD_{600nm}$ = 5 was harvested by centrifugation (5000 x $g$, 10 min) and washed with 500 µL of cold water. The supernatant was discarded before adding 400 µL of cold 5% TCA (Trichloroacetic acid). Glass beads were added until reaching the meniscus. Yeasts were broken using MP Biomedicals FastPrep-24 during 4 × 20 s, chilling tubes on ice between vortexing to keep cells chilled. The supernatant was collected and beads were washed three times with 400 µL cold TCA. All supernatants were pooled and chilled on ice for 30 min. Proteins were pelleted by centrifugation at 16,000 x $g$ for 30 min at 4 °C. Pellets were resuspended with 100 µL of 100 mM Tris-HCl pH 7.5.

**Immunodetection of UCP1.** To ensure the equivalent expression level among all mutants and all biological replicates, 10 µL (equivalent to $OD_{600nm}$ = 0.5) TCA extracts of total yeast or 2 µg of mitochondrial proteins were loaded on a 12% SDS-PAGE and transferred onto a nitrocellulose membrane (GE Healthcare). UCP1 was detected using a mouse anti-pentahistidine tag:HRP 0.2 µL/mL (Bio-RAD, catalogue number: MCA5995P, clone: ABD 2.2.20). VDAC1 Porin was used as a mitochondrial loading control and detected by a mouse anti-VDAC1 0.1 µL/mL (Invitrogen, catalogue number: 459500, clone: 16G9E6BC4) and a goat anti-mouse tag: HRP 0.1 µL/mL (Promega, catalogue number: W4021, polyclonal). Uncropped scans of all blots are provided as a Source Data file.

**Respiration measurements.** To assess UCP1 activity, $O_2$ consumption measurements were performed using an Oroboros instrument at 28 °C under 750 rpm agitation. Spheroplasts (2.5 mg/mL in respiration buffer with 250 µg/mL nystatin) were permeabilized for 15 min at 28 °C. Before the UCP1 activity measurement, the respiratory control ratio (RCR) was measured by successive addition of NADH (3.125 mM), ADP (0.625 mM), oligomycin (1 µM), and carbonyl cyanide m-chlorophenyl hydrazone (CCCP, 10 µM).

FFAs trigger proton leak activity in several mitochondrial carriers, especially in AAC[14,15], which is highly expressed in yeast mitochondria. We searched for an LA concentration that selectively stimulates UCP1-dependent uncoupling of respiration with no effect on AAC. Above 120 micromolar concentration, LA stimulates uncoupling of respiration in control yeast spheroplasts. The addition of CATR inhibits this increase in respiration showing that the main target of LA is yeast AAC (Supplementary Fig. 9a). The 60 micromolar concentration corresponding to the ratio LA/BSA = 4 was selected to stimulate UCP1 activity in yeast recombinant spheroplasts. At this concentration of LA, CATR does not affect respiration, showing that endogenous yeast AAC does not contribute to the LA-induced proton leak (Supplementary Fig. 9). As previously observed[18], this ratio LA/BSA = 4, does not affect control spheroplasts' respiration. In contrast, it significantly increases the respiration of spheroplasts harboring UCP1 to 55% of the maximal respiration rate ($VO_{2max}$) (Supplementary Figs. 10 and 3d).

The RCR was calculated by dividing the mean $O_2$-ADP flux by the mean $O_2$-oligomycin flux. RCR determination was performed with each independent yeast culture. UCP1 activity and inhibition were measured according to the following sequence: oligomycin (1 µM), NADH (3.125 mM), lauric acid or palmitic acid (60 µM, corresponding to a ratio fatty acid/BSA = 4), GDP or GTP (1.25 mM), CCCP (10 µM). GDP was chosen over ADP or ATP to minimize artefacts due to transport by endogenous AAC in yeast. Mean $O_2$ fluxes were calculated from respiration curves with DatLab Oroboros software after each addition with a time window of 1 min. Data are presented as mean ± SEM (Standard Error of the Mean). Measurements were taken from at least three distinct biological samples. For interpretation purposes, data were normalized by $O_2$-CCCP flux ($F_{CCCP}$) after removing $O_2$-NADH or by calculating the inhibition rate with $1 - \frac{F_{GDP} - F_{NADH}}{F_{AL} - F_{NADH}}$. Significant differences between mutants are assessed based on a One-way ANOVA and Donnett's multiple comparison test with rUCP1-WT. Significant differences between AL and AP or GDP and GTP are assessed with a two-tailed t test. ns = not significant, * = $p$-value ≤ 0.05, ** = $p$ ≤ 0.01, *** = $p$-value ≤ 0.001, **** = $p$-value ≤ 0.0001. Boxes of the boxplots indicates the first quartile, the median, and the third quartile. The whisker length is 1.5 times the inter-quartile range.

## Reporting summary

Further information on research design is available in the Nature Portfolio Reporting Summary linked to this article.

## Data availability

All data necessary for reproducing simulation results are available through Zenodo (https://doi.org/10.5281/zenodo.7698270)[58]. PDB codes of previously published structures used in this study are: 2C3E (*Bos taurus* mitochondrial ADP/ATP carrier) 2LCK (*Mus musculus* mitochondrial UCP2) 4C9Q (*Saccharomyces cerevisiae* ADP/ATP carrier isoform 3 inhibited by carboxyatractyloside) 6GCI (*Thermothelomyces thermophilus* mitochondrial ADP/ATP carrier) 7W5Z (*Tetrahymena thermophila* mitochondrial complex IV) chain M2 (Uniprot entry Q23M99). Source data are provided with this paper.

## Code availability

All necessary software for reproducing simulation results is freely available. NAMD can be downloaded from the NAMD home page (http://www.ks.uiuc.edu/Research/namd), and the ABMD script from the Colvars repository (https://github.com/Colvars/colvars).

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

## Acknowledgements

The authors acknowledge support from ANR through LABEX DYNAMO (ANR-11-LABX-0011) and Equipex CACSICE (ANR-11-EQPX-0008) for the visualization wall, and from the Laboratoire International Associé of CNRS and UIUC. We are grateful to François Dehez and Christophe Chipot for stimulating discussions and Céline Ransy for help with respiration assays. AG and CL are supported by a French Ministry Higher Education, Research and Innovation PhD fellowship. This work was performed using HPC resources from GENCI-CINES (Grant A0100710760) and the LBT-HPC cluster managed by Geoffrey Letessier.

## Author contributions

A.G., C.L., S.M., M.S.P., F.B., J.H., and B.M. designed the experiments and analyzed the data. A.G., C.L., S.M., M.S.P., and D.C.M. performed the experiments. A.G., C.L., J.H., and B.M. wrote the paper with contributions from all authors.

## Competing interests

The authors declare no competing interests.
