## [Peer Review File · Nature Communications]

Molecular determinants of inhibition of UCP1-mediated respiratory uncouplingReviewer #1 (Remarks to the Author):

“Molecular determinants of inhibition of UCP1-mediated respiratory uncoupling” by Gagelin, Largeau et al.

In this paper, the authors use a combination of molecular dynamics studies, sequence information, and functional assays of single replacement mutants to probe the molecular mechanism of nucleotide binding to rat uncoupling protein (UCP1). This is an important aspect of the function of purine nucleotides in the inhibition of heat generation in brown adipose tissue. The work revisits some earlier claims (e.g salt bridge formation between E191-R92), but also provides new molecular detail to GDP binding for instance the involvement of FIW88. In particular, new insights are also obtained in the nucleotide specificity, e.g with regards to UDP binding, although it is somewhat partial.

The alignment in Figure 1 should contain the Tetrahymena sequence of 7W5Z. It is clear that this sequence, which most closely resembles the oxoglutarate carrier, is actually more closely related to UCP1 than AAC1, as UCP1 and the oxoglutarate carrier belong to the same clade in the SLC25 family, according to Palmieri and colleagues. The alphafold model is largely based on AAC1, so it is entirely possible that the Tetrahymena model is the closest to the real UCP1, and in agreement it shows the least amount of splaying of the three models. If so, it is not clear whether any of the following studies were carried out with this model. More importantly whether the same conclusions would have been reached, if they were. If not, it would be prudent to repeat the MD binding studies with the Tetrahymena model to see if they are consistent.

“In comparison, AAC structures and UCP1 models based thereon exhibit little or no permeability, and GDP binding further decreases the permeability of the C-state UCP1 model.” This is an ambiguous statement; there either permeability or not. If the C-state does open on occasion then it would be good to show a measure of it. Still, this would be odd, as it would create a proton leak.

Residue E191 was believed to be responsible for the pH-dependency of purine binding. Were simulations carried out with E191 being negatively charged or E191 being protonated and thus neutral? This is crucial with regards to GDP binding as well as the salt bridge formation with R92. Should be said clearly. Were controls done?

Line 276-294 Some good comments are made contrasting AAC and UCP1 with regards to nucleotide binding in relation to transport and inhibitory binding, respectively, but it fails to make the most important point (in addition to some errors). Transport can only occur if the substrate binding allows the gates to open and close on either side of the binding site. This is satisfied in AAC, as the substrate binds centrally, whereas in UCP1 it binds in an upright position, preventing closure of the gate. Reading Mavridou et al, it seems that the base does not flip (as claimed in line 288), as there is only one binding site for adenosine, which is located centrally and accessible in both states (see figure 4, but also Ruprecht 2019). The phosphate groups of ADP/ATP do all of the flipping in the transport mechanism of AAC. In UCP1, the base binding prevents the closure of the cytoplasmic gate (investigate this point more thoroughly) and the phosphates do not flip at all and are severely cross-linked by binding to several positively charged residues (triplet and network). GDP binding to UCP1 is more akin to carboxyatractyloside binding to AAC, which also binds to the central site in an upright (orthogonal) position and to the matrix network, preventing opening (see Pebay-Peyroula et al and Ruprecht et al). Thus, the key point is that inhibitor binding prevents the opening and closing of the two gates, whereas substrate binding does not.

Minor corrections

Line 59 to occur.[5] -> to occur [5].

Line 66 by AAC but not UCP1 -> by AAC but not by UCP1

Line 93 Because AAC sequences -> Since AAC sequences

Figure 1A non-transmembrane helix -> matrix helix (h12, h34, h56), as defined by Pebay-Peyroula et al, 2003.

Line 118 In contrast, binding trajectories to the C-state conformation show much more structure. This is an ambiguous comment. Maybe you mean: lead to a larger number of possible conformations.

Line 135 In the M-state conformation, 1 μ s unbiased relaxation trajectories confirm the absence of preferential orientation and broad distribution of positions of GDP within the cavity. This sentence is ambiguous and seems to be in contrast to line 116. Needs a better formulation.

Line 167 both amino acids' function controlling UCP1 inhibition by nucleotides -> the function of both amino acids in controlling UCP1 inhibition by nucleotides.

Line 266-267 Not aware of any conclusive evidence that shows the binding site for fatty acids, and thus it is unclear whether GDP binds competitively to the same site.

Reviewer #2 (Remarks to the Author):

Gagelin, Largeau et al. present their study into the inhibition of UCP1-mediated respiratory uncoupling, applying molecular dynamics simulations and mitochondrial respiration assays.

The manuscript is, on the whole, well-written and easy to follow, with excellent illustrative items. I have a few points below, which I hope will aid the authors to improve their manuscript:

1. As the molecular simulations are based on homology models of the C- and M-states of UCP1, it would be useful to the reader to indicate the quality of the structures; especially as the sequence identity is only 20 % (line 93). I recommend providing a supplementary figure that details the structural assessment of the models, e.g.

<https://swissmodel.expasy.org/assess>, and quote within the manuscript (perhaps also with a SI Table) the Molprobit and/or QMEAN scores. These should be compared with the X-ray, EM, NMR and AlphaFold structures of the related AAC/UCP2 proteins.

2. Line 76 - 'High-resolution crystal structures exist for the ATP/ADP Carrier (AAC), and AAC, like UCP1, has been shown to exhibit proton transport activity and to contribute to fatty-acid-dependent uncoupling of respiration' took me a number of re-reads to fully grasp the subject of the second-half of sentence. Perhaps split in two: 'High-resolution crystal structures exist for the ATP/ADP Carrier (AAC). AACs, like UCP1, have been shown to exhibit proton transport activity and to contribute to fatty-acid-dependent uncoupling of respiration'. The problem of AAC being both a protein and a family name.

3. Figure 1 text - UCP2 is an NMR structure. Also note in the text what the C-state model of UCP1 was based on (i.e. PDB id). This should be stated in both (a) and (b) to avoid confusion. As the AlphaFold model is from AFDB a link to the model should be provided or simply the UniProtID. (d) Looks to me that the AF model is the core structure that both the 7W5Z and AAC model converge towards over 300-ns. Again - not sure why AAC is not afforded a PDB id but 7W5Z is.

4. There is inconsistency, in a number of places, in the hyphenation of ns. Similarly, Z-axis is both z-axis and Z axis. C- and M-state also have various hyphenations throughout.

5. In Figure 2b the black sphere that denotes centre of the protein rather dominates the image. Could it be made transparent? Or redrawn in Inkscape or equivalent as an unfilled circle?

6. In Figure 3b, the R is missing from the double-mutant name.

7. In Figure 4a - are the error bars only shown for the 'other contacts'?

8. Line 260: Molecular dynamics simulations on AAC -> Molecular dynamics simulations of AAC

Reviewer #3 Attachment on the following page

Molecular determinants of inhibition of UCP1-mediated 1 respiratory uncoupling

Comments to the Author:

Gagelin et al analyzed the structural determinants of the inhibition of the brown and beige fat uncoupling protein, UCP1, by GDP. To do so, the authors mainly use molecular dynamics simulations and mitochondrial respiration. UCP1 is a member of the large mitochondrial family of SLC25 transporters, responsible for proton transport across the inner mitochondrial membrane and heat production by mitochondria. It is accepted in the mitochondrial field that members of the SLC25 family have broadly similar structure (6 transmembrane domains) with a transport mechanism that adopt a c-state conformation and switch to the m-state conformation to transport substrates. The authors constructed three-dimensional structures of UCP1 based on the two well-established crystal structures of the ADP/ATP carrier (AAC), one in the c-state and the other in the m-state. They use the c-state protein structure of bovine AAC1 (stabilized with carboxyatractyloside - CATR) and the m-state protein structure of *Thermothelomyces thermophila* AAC (stabilized with bongkrekeic acid - BKA). Once the UCP1 models were constructed, the authors introduced GDP, a specific inhibitor of UCP1, either on the c- or m-state of the protein. They found that GDP positions in the common substrate binding site in the vertical orientation, where the base moiety interacts with a pair of charged residues (R92/E191). E191 interacts with purine but not pyrimidines, which would explain the nucleotide specificity in UCP1 inhibition. They also found a triplet of uncharged residues involved in hydrophobic contacts with GDP. Based on the simulations, they performed site-directed mutagenesis of rat UCP1 expressed in yeast of the key residues and studied the impact of uncoupled mitochondrial respiration. Site-directed mutagenesis of I187 or W281 to alanine enhances lauric acid-induced uncoupling activity of UCP1 and partially suppresses GDP-mediated inhibition of UCP1 activity in yeast spheroplasts. Interestingly, they identified mutants with higher proton transport potential that become partially or nearly insensitive to GDP inhibition.

The results are well explained, and the manuscript is enjoyable to read. However, the reviewer is not entirely convinced and would like several points to be addressed. Please see below:

Major issues:

- 1- Since fatty acids (FAs) are the substrate for UCP1, simulations with FAs will allow to understand how the salt bridge behaves in the c-state and how the substrate binding site might rearrange. Indeed, substrates are also likely to be able to impact this salt bridge. Long-chain fatty acids such as palmitic acid or arachidonic acid should be preferable. The mutants apparently described in the manuscript with mitochondrial respiration are quite interesting and help answer some questions that the simulations might not show. However, it is not clear whether GDP should destabilize the AAC salt bridge when FAs are present. Once FAs interact with UCP1, it is also unclear how GDP will position itself in the cavity. This is related to the competition model between UCP1 and GDP. Simulations of FAs and GDP together would be elegant. Perhaps, sequentially, GDP simulations can be done after UCP1 simulations with FAs.
- 2- Simulations have well shown that only purines can interact with UCP1 and have identified the key residues E191, W281, or Q85 responsible for the selectivity of UCP1 inhibition by purine nucleotides. However, the simulations do not really highlight the importance of the number of phosphates for the positioning of nucleotides in UCP1. It is known that UCP1 is inhibited by GDP and ADP, but also by ATP and GTP. Simulations with GTP or ATP will help to better understand the interactions of phosphates with the UCP1 cavity. Are GDP-insensitive UCP1 mutants also GTP-insensitive in mitochondrial respiration?

- 3- The authors described the interaction between GDP and phospholipids (lines 190-191). It would be interesting to study whether the FA simulations also describe an interaction between UCP1 and phospholipids when FAs are present. It would also be interesting to study whether phospholipids position themselves in the FA binding site when they are absent.
- 4- Mitochondrial respiration experiments were performed with lauric acid, a medium chain fatty acid. UCP1 shows stronger interactions with long chain fatty acids such as palmitic acid or arachidonic acid. It would be important to repeat these experiments, or at least key experiments with long chain fatty acids to rule out the possibility of inadequate interaction of lauric acid with UCP1.

Minor issues:

- 1- It is stated in the text that AAC has been successfully used to simulate another uncoupling protein structure, UCP2. The reviewer understands that this provides a rationale for using the AAC structure for another uncoupling protein, here UCP1. However, several works argue against UCP2 being a protein capable of transporting H⁺. Thus, successfully simulating UCP2 in the concept that it can transport H⁺ may not be a benchmark. It is important to rephrase this point (line 83). As set in the text, the authors performed simulations based on AACs not only because the structures of AACs are well established but also because only UCP1 and AACs have been confirmed to transport H⁺ in the inner mitochondrial membrane.
- 2- It says line 113 "consistent with experimental evidence that UCP1 is accessible to ATP on both sides". The cited article is based on lipid bilayer experiments that could produce questionable results. Please expand and add references using alternative experiments.

We thank the Reviewers for their in-depth evaluation of the manuscript and their constructive comments. As requested, we have performed additional MD simulations as well as functional respiration experiments to answer the Reviewers' questions and comments. We have shortened the abstract to comply with the journal guidelines. Highlighted changes are provided in a separate file.

Reviewer #1 (Remarks to the Author):

"Molecular determinants of inhibition of UCP1-mediated respiratory uncoupling" by Gagelin, Largeau et al.

In this paper, the authors use a combination of molecular dynamics studies, sequence information, and functional assays of single replacement mutants to probe the molecular mechanism of nucleotide binding to rat uncoupling protein (UCP1). This is an important aspect of the function of purine nucleotides in the inhibition of heat generation in brown adipose tissue. The work revisits some earlier claims (e.g salt bridge formation between E191-R92), but also provides new molecular detail to GDP binding for instance the involvement of FIW88. In particular, new insights are also obtained in the nucleotide specificity, e.g with regards to UDP binding, although it is somewhat partial.

The alignment in Figure 1 should contain the *Tetrahymena* sequence of 7W5Z. It is clear that this sequence, which most closely resembles the oxoglutarate carrier, is actually more closely related to UCP1 than AAC1, as UCP1 and the oxoglutarate carrier belong to the same clade in the SLC25 family, according to Palmieri and colleagues. The alphafold model is largely based on AAC1, so it is entirely possible that the *Tetrahymena* model is the closest to the real UCP1, and in agreement it shows the least amount of splaying of the three models. If so, it is not clear whether any of the following studies were carried out with this model. More importantly whether the same conclusions would have been reached, if they were. If not, it would be prudent to repeat the MD binding studies with the *Tetrahymena* model to see if they are consistent.

*We agree with the Reviewer that UCP1 and the oxoglutarate carrier belong to the same clade and that the 7W5Z sequence resembles that of the oxoglutarate carrier. 7W5Z sequence has been added to the alignment of Figure 1a. We have extended our investigation of this model by MD simulations, to 1- μ s trajectories of the apo UCP1-7W5Z models. Analysis of these trajectories confirm that these models are close in structure and behavior to our C-state UCP1-AAC models, and that they further converge during simulations. In particular, the structure of residues interacting with nucleotides in the common substrate binding sites is very similar between the two models (RMSD below 2 Angstroms, see Figure S1). Our conclusion is that both of those templates yield equivalent models for the purposes of this study. However, a few SLC25 sequences from *Tetrahymena*, including the carrier found in 7W5Z, seem to have lost the key salt bridge of the cytoplasmic salt bridge network (see new alignment in revised Figure 1). We therefore prefer to keep the AAC model as reference, in which both salt bridge networks are fully conserved, as for UCP1 and almost all oxoglutarate carrier sequences. Figure 1d in its new version shows that the 7W5Z based model also relaxes to the same opening profile as the other models.*

We have done the following modifications in the manuscript:

Line 99

We have extended our investigation of the 7W5Z-based model by MD simulations, to 1 μ s trajectories of apo UCP1. Analysis of these trajectories confirms that these models are close in structure and behavior to our C-state UCP1-AAC models, and that they further converge during simulations. In particular, the structure of residues interacting with nucleotides in the common substrate binding sites is very similar between the two sets of homology models (RMSD below 2 Å, see Figure S1). Our conclusion is that both of those templates yield equivalent models for the purposes of this study. However, the carrier found in 7W5Z exhibits sequence divergence at the cytoplasmic salt-bridge network (Figure 1).

"In comparison, AAC structures and UCP1 models based thereon exhibit little or no permeability, and GDP binding further decreases the permeability of the C-state UCP1 model." This is an ambiguous statement; there either permeability or not. If the C-state does open on occasion then it would be good to show a measure of it. Still, this would be odd, as it would create a proton leak.

The Reviewer is right that in the mentioned sentence, we seemed to imply a direct link between water permeability and proton transport. The main purpose of measuring water permeability was to assess the quality of our models as we have done in Zoonens et al. (2013) to assess the NMR based UCP2 models. A large water channel with multiple water columns as observed in the UCP2 model would certainly create a proton leak. However, we have no evidence of a link between the low levels of water permeability observed by MD and proton transport and this is a subject of a separate study. Water permeability does not come from the central cavity and the matrix side network of salt bridges remains formed in all simulations of C-state models. We observed, in one replica, and in a reversible manner, water leak through alpha-helices on the matrix side. Our new extended simulations of the UCP1-7W5Z model match the behavior of the low-permeability UCP1-AAC replicas, confirming that the isolated water permeation event should not be given more statistical weight.

In the revised version of the paper, we have included the water permeability metric of 7W5Z-based models in the revision (Figure S2), and amended the sentence above as:

Line 108

Previous version:

The NMR structure of UCP2 is known to be fully open and water-permeable [17], and our model of UCP1 follows the same trend. In comparison, AAC structures and UCP1 models based thereon exhibit little or no permeability, and GDP binding further decreases the permeability of the C-state UCP1 model.

New version:

The NMR structure of UCP2 is known to be fully open and water-permeable [17], and the related UCP1 model (UCP1-NMR) follows the same trend, which is expected to be incompatible with controlled proton transport. In comparison, AAC structures and UCP1 models based thereon or on 7W5Z exhibit little or no water permeability.

Residue E191 was believed to be responsible for the pH-dependency of purine binding. Were simulations carried out with E191 being negatively charged or E191 being protonated and thus neutral? This is crucial with regards to GDP binding as well as the salt bridge formation with R92. Should be said clearly. Were controls done?

The Klingenberg group has indeed proposed that E191 is responsible for the pH-dependency of purine binding to UCP1 in vitro. When preparing simulations pKas were predicted using the Propka3 software, which predicts a pKa of 5.91 for E191 in the UCP1-AAC C-state model. Accordingly, residue E191 was deprotonated in all simulations, as indicated I. 373. The predicted pKa decreased to around 4 after 1 μ s of simulations, which confirms our initial choice of protonation state. In this study, we have focused on molecular determinants that control UCP1-dependent uncoupling of respiration in a physiological range of pH. We have therefore not varied the pH computationally to stay close to the pH conditions defined by respiration assay on spheroplasts.

Line 276-294 Some good comments are made contrasting AAC and UCP1 with regards to nucleotide binding in relation to transport and inhibitory binding, respectively, but it fails to make the most important point (in addition to some errors). Transport can only occur if the substrate binding allows the gates to open and close on either side of the binding site. This is satisfied in AAC, as the substrate binds centrally, whereas in UCP1 it binds in an upright position, preventing closure of the gate. Reading Mavridou et al, it seems that the base does not flip (as claimed in line 288), as there is only one binding site for adenosine, which is located centrally and accessible in both states (see figure 4, but also Ruprecht 2019). The phosphate groups of ADP/ATP do all of the flipping in the transport mechanism of AAC. In UCP1, the base binding prevents the closure of the cytoplasmic gate (investigate this point more thoroughly) and the phosphates do not flip at all and are severely cross-linked by binding to several positively charged residues (triplet and network). GDP binding to UCP1 is more akin to carboxyatractyloside binding to AAC, which also binds to the central site in an upright (orthogonal) position and to the matrix network, preventing opening (see Pebay-Peyroula et al and Ruprecht et al). Thus, the key point is that inhibitor binding prevents the opening and closing of the two gates, whereas substrate binding does not.

We agree with the Reviewer that in the case of substrate transport, the key mechanism of inhibition is binding of the inhibitor, preventing the transition between C- and M-State. However,

in the case of UCP1, we have no evidence of substrate transport mediated by a state transition. Therefore, it is difficult to compare the AAC transition state inhibitors CATR or BA to the proton transport inhibitors of UCP. As pointed out by the Reviewer, we had not correctly quoted the model proposed in reference 36. We have now rectified the sentence on line 290 as follows:

Line 309

*In a recent conceptual model of nucleotide translocation by AAC, the small size of G199 leaves a gap that can accommodate the base moiety (Figure 2a in ref. [5]). This would allow the **base** to flip to the matrix side, which is expected to be necessary for transport [36].*

*In a recent conceptual model of nucleotide translocation by AAC, the small size of G199 leaves a gap that can accommodate the base moiety (Figure 2a in ref. [5]). This would allow the **phosphate groups** to flip to the matrix side, which is expected to be necessary for transport [36].*

Minor corrections

Line 59 to occur.[5] -> to occur [5].

Line 66 by AAC but not UCP1 -> by AAC but not by UCP1

Line 93 Because AAC sequences -> Since AAC sequences

Figure 1A non-transmembrane helix -> matrix helix (h12, h34, h56), as defined by Pebay-Peyroula et al, 2003.

These corrections have been made.

Line 118 In contrast, binding trajectories to the C-state conformation show much more structure. This is an ambiguous comment. Maybe you mean: lead to a larger number of possible conformations.

We have clarified this sentence as follows:

Line 127

*In contrast, binding trajectories to the C-state conformation **show much more structure.***

*In contrast, binding trajectories to the C-state conformation **are more constrained and fall into well-separated categories.***

Line 135 In the M-state conformation, 1 μ s unbiased relaxation trajectories confirm the absence of preferential orientation and broad distribution of positions of GDP within the cavity. This sentence is ambiguous and seems to be in contrast to line 116. Needs a better formulation.

Line 143

*In trajectories where GDP binds in the opposite orientation, spontaneous unbinding is observed within 1 μ s. In the M-state conformation, 1 μ s unbiased relaxation trajectories confirm the absence of preferential orientation **and broad distribution of positions of GDP** within the cavity.*

*In trajectories where GDP binds in the opposite orientation, spontaneous unbinding is observed within 1 μ s. In the M-state conformation, 1 μ s unbiased relaxation trajectories confirm the absence of preferential orientation **of the base moiety** within the cavity.*

Line 167 both amino acids' function controlling UCP1 inhibition by nucleotides -> the function of both amino acids in controlling UCP1 inhibition by nucleotides.

This correction has been made.

Line 266-267 Not aware of any conclusive evidence that shows the binding site for fatty acids, and thus it is unclear whether GDP binds competitively to the same site.

This point is also addressed by Reviewer 3. We did some simulations of UCP1 with FAs with and without GDP, but we did not observe any significant effect of FAs on the interactions between the nucleotide and the protein. A specific FA binding site has been found by MD simulation in AAC (Bertholet et al. 2022), which, so far, has not been found within the UCP1 cavity either by us in this study or other laboratories. We have therefore too little information about an FA binding site to perform a reliable MD competition assay between FA and nucleotides, as pertinently suggested by the Reviewer.

For the same reason, and as suggested by Reviewer 3, we removed from the Discussion the following sentence related to competition between FAs and nucleotides:

Line 292

Given that both I187A and W281A single mutations enhance UCP1 uncoupling activity with a significant loss of inhibition by nucleotides (Figure 5b), and despite sequence and asymmetric differences between the two triplets (TYG in AAC, FIW88 in UCP1), these data support a competition model for both carriers between FA and the base moiety of purine nucleotides [32]. At the level of FIW88 triplet within the cavity of UCP1, LA and nucleotide would compete for interaction with both I187 and W281 residues.

Reviewer #2 (Remarks to the Author):

Gagelin, Largeau et al. present their study into the inhibition of UCP1-mediated respiratory uncoupling, applying molecular dynamics simulations and mitochondrial respiration assays.

The manuscript is, on the whole, well-written and easy to follow, with excellent illustrative items. I have a few points below, which I hope will aid the authors to improve their manuscript:

1. As the molecular simulations are based on homology models of the C- and M-states of UCP1, it would be useful to the reader to indicate the quality of the structures; especially as the sequence identity is only 20 % (line 93). I recommend providing a supplementary figure that details the structural assessment of the models, e.g. <https://swissmodel.expasy.org/assess>, and quote within the manuscript (perhaps also with a SI Table) the Molprobity and/or QMEAN scores. These should be compared with the X-ray, EM, NMR and AlphaFold structures of the related AAC/UCP2 proteins.

We have added the QMEAN scores for all structures as Supplementary Figure S4, and the following commentary:

Line 112

Finally, the QMEAN6 [25] structure quality score was computed on the models at different stages of the simulation and compared to the experimental structures of AAC as reference (Figure S3). The C-state and M-state UCP1-AAC models have scores between those of the corresponding AAC crystal structures, indicating that the homology modeling did not significantly deteriorate the overall quality of the structures.

2. Line 76 – ‘High-resolution crystal structures exist for the ATP/ADP Carrier (AAC), and AAC, like UCP1, has been shown to exhibit proton transport activity and to contribute to fatty-acid-dependent uncoupling of respiration’ took me a number of re-reads to fully grasp the subject of the second-half of sentence. Perhaps split in two: ‘High-resolution crystal structures exist for the ATP/ADP Carrier (AAC). AACs, like UCP1, have been shown to exhibit proton transport activity and to contribute to fatty-acid-dependent uncoupling of respiration’. The problem of AAC being both a protein and a family name.

Thank you for this remark which has drawn our attention to potential confusion created by this sentence. For clarity, we have amended it in the following way:

Line 75

High-resolution crystal structures exist for the ATP/ADP Carrier (AAC), and AAC, like UCP1, has been shown to exhibit proton transport activity and to contribute to fatty-acid-dependent uncoupling of respiration.

High-resolution crystal structures exist for the ATP/ADP Carrier (AAC), which has also been shown to exhibit proton transport activity and to contribute to fatty-acid-dependent uncoupling of respiration.

3. Figure 1 text - UCP2 is an NMR structure.

Thank you for noticing this error.

Also note in the text what the C-state model of UCP1 was based on (i.e. PDB id). This should be stated in both (a) and (b) to avoid confusion.

We have added PDB code 2C3E to the figure caption.

As the AlphaFold model is from AFDB a link to the model should be provided or simply the UniProtID.

Line 359

The UCP1-AlphaFold model was retrieved from the AlphaFold Protein Structure Database.

The UCP1-AlphaFold model was retrieved from the AlphaFold Protein Structure Database (UniProt P04633).

(d) Looks to me that the AF model is the core structure that both the 7W5Z and AAC model converge towards over 300-ns.

To make this easier to read, we have overlaid the starting and ending profiles onto the same graph. One can now see that the homology models converge to an intermediate profile, and that the AlphaFold model is not as typical.

Line 96

A global comparison of these models indicates variations in the splay of helices on the cytoplasmic side (Figure 1c), which tends to decrease upon physical relaxation in an all-atom MD simulation (Figure 1d), converging towards **the least splayed variant, based on cryo-EM structure 7W5Z**.

A global comparison of these models indicates variations in the splay of helices on the cytoplasmic side (Figure 1c), which tends to decrease upon physical relaxation in an all-atom MD simulation (Figure 1d), converging towards **an intermediate configuration between UCP1-7W5Z and UCP1-AAC**.

Again – not sure why AAC is not afforded a PDB id but 7W5Z is.

We would rather not use PDB identifiers in our model names as they are not intuitive to most readers. We defaulted to using the PDB ID in this case because the putative carrier resolved in structure 7W5Z that we use as a template is uncharacterized, so that we had no reliable name for it.

4. There is inconsistency, in a number of places, in the hyphenation of ns. Similarly, Z-axis is both z-axis and Z axis. C- and M-state also have various hyphenations throughout.

Thank you, we have addressed this.

5. In Figure 2b the black sphere that denotes centre of the protein rather dominates the image. Could it be made transparent? Or redrawn in Inkscape or equivalent as an unfilled circle?

We had a collective discussion about this comment and we would like to keep the center of the protein clearly visible. We hope this is acceptable for the Reviewer.

6. In Figure 3b, the R is missing from the double-mutant name.

Figure 3d has been corrected.

7. In Figure 4a – are the error bars only shown for the 'other contacts'?

Error bars in that figure refer to the total contacts, we changed the legend to make it less ambiguous. Note that the error bars for triplet RRR84 are so small as to be barely visible.

Reviewer #3 (Remarks to the Author):

Gagelin et al analyzed the structural determinants of the inhibition of the brown and beige fat uncoupling protein, UCP1, by GDP. To do so, the authors mainly use molecular dynamics simulations and mitochondrial respiration. UCP1 is a member of the large mitochondrial family of SLC25 transporters, responsible for proton transport across the inner mitochondrial membrane and heat production by mitochondria. It is accepted in the mitochondrial field that members of the SLC25 family have broadly similar structure (6 transmembrane domains) with a transport mechanism that adopt a c-state conformation and switch to the m-state conformation to transport substrates. The authors constructed three-dimensional structures of UCP1 based on the two well-established crystal structures of the ADP/ATP carrier (AAC), one in the c-state and the other in the m-state. They use the c-state protein structure of bovine AAC1 (stabilized with carboxyatractyloside - CATR) and the m-state protein structure of *Thermothelomyces thermophila* AAC (stabilized with bongkreikic acid - BKA). Once the UCP1 models were constructed, the authors introduced GDP, a specific inhibitor of UCP1, either on the c- or m-state of the protein. They found that GDP positions in the common substrate binding site in the vertical orientation, where the base moiety interacts with a pair of charged residues (R92/E191). E191 interacts with purine but not pyrimidines, which would explain the nucleotide specificity in UCP1 inhibition. They also found a triplet of uncharged residues involved in hydrophobic contacts with GDP. Based on the simulations, they performed site-directed mutagenesis of rat UCP1 expressed in yeast of the key residues and studied the impact of uncoupled mitochondrial respiration. Site-directed mutagenesis of I187 or W281 to alanine enhances lauric acid-induced uncoupling activity of UCP1 and partially suppresses GDP-mediated inhibition of UCP1 activity in yeast spheroplasts. Interestingly, they identified mutants with higher proton transport potential that become partially or nearly insensitive to GDP inhibition.

The results are well explained, and the manuscript is enjoyable to read. However, the Reviewer is not entirely convinced and would like several points to be addressed. Please see below:

Major issues:

1 - Since fatty acids (FAs) are the substrate for UCP1, simulations with FAs will allow to understand how the salt bridge behaves in the c-state and how the substrate binding site might rearrange. Indeed, substrates are also likely to be able to impact this salt bridge. Long-chain fatty acids such as palmitic acid or arachidonic acid should be preferable.

See issue number 4 below for our reply on the question of long-chain fatty acids.

The mutants apparently described in the manuscript with mitochondrial respiration are quite interesting and help answer some questions that the simulations might not show. However, it is not clear whether GDP should destabilize the AAC salt bridge when FAs are present. Once FAs interact with UCP1, it is also unclear how GDP will position itself in the cavity. This is related to the competition model between UCP1 and GDP. Simulations of FAs and GDP together would be elegant. Perhaps, sequentially, GDP simulations can be done after UCP1 simulations with FAs.

We very much agree with this suggestion. We did some simulations of UCP1 with FAs with and without GDP, but we did not observe any significant effect of FAs on the interactions between the nucleotide and the protein. A specific FA binding site has been found by MD simulation in AAC (Bertholet et al. 2022), which, so far, has not been found within the UCP1 cavity either by us in this study or other laboratories. We have therefore too little information about an FA binding site to perform a reliable MD competition assay between FA and nucleotides, as pertinently suggested by the Reviewer.

For the same reason, and as suggested by Reviewer 1, we removed from the Discussion the following sentence related to competition between FAs and nucleotides:

Line 292

Given that both I187A and W281A single mutations enhance UCP1 uncoupling activity with a significant loss of inhibition by nucleotides (Figure 5b), and despite sequence and asymmetric differences between the two triplets (TYG in AAC, FIW88 in UCP1), these data support a competition model for both carriers between FA and the base moiety of purine nucleotides [32]. At the level of FIW88 triplet within the cavity of UCP1, LA and nucleotide would compete for interaction with both I187 and W281 residues.

2 - Simulations have well shown that only purines can interact with UCP1 and have identified the key residues E191, W281, or Q85 responsible for the selectivity of UCP1 inhibition by purine nucleotides. However, the simulations do not really highlight the importance of the number of phosphates for the

positioning of nucleotides in UCP1. It is known that UCP1 is inhibited by GDP and ADP, but also by ATP and GTP. Simulations with GTP or ATP will help to better understand the interactions of phosphates with the UCP1 cavity.

We agree with the Reviewer and as suggested, we have performed simulations with GTP and compared by MD simulation and experimentally, UCP1 inhibition with GDP and GTP. These results appear in an additional Figure S9 and in the "Molecular bases for nucleotide specificity of UCP1 inhibition" section renamed "UCP1 inhibition is insensitive to FA chain length and to the number of nucleotide phosphates".

The wording of the abstract mentioning GDP and lauric acid has been made more general to include GTP and palmitic acid, and we added the following text to describe these new results.

Line 231

Long-chain fatty acids have been shown to better activate UCP1 than medium-chain FA [26]. In order to rule out possible effects of FA chain length on our results, we performed respiration experiments with palmitic acid (C16), a widely used activator of UCP1. We find slightly higher activation by palmitic acid (Figure S9d), as described in [26]. Despite this increase in activation, rUCP1-WT or variants show similar phenotypes of inhibition by GDP (Figure S9a, b).

To assess the importance of phosphates in UCP1-nucleotide interactions, we measured GTP inhibition of UCP1 after activation by lauric or palmitic acid. We find almost no significant difference between inhibition by GDP and GTP, after activation with either type of fatty acid, for either the wild-type or the variants documented above. In the case of rUCP1-WT, inhibition by GTP in response to palmitic acid as opposed to lauric acid is smaller by 20 percentage points. We also investigated UCP1-GTP interactions by MD simulations (Figure S9c). GTP and GDP exhibit similar contacts with UCP1, especially between di/triphosphate moiety and triplet RRR84 at the bottom of the cavity. Contact times with key residues identified in the GDP simulation were mainly unchanged, showing the specificity of interactions in this region; the main difference is a reduced interaction time with Q85.

These new simulations of UCP1 with GTP highlighted a low contact rate between Q85 and the nucleotide. Thus, we have amended the text as following:

Line 249

As expected, UDP and GDP show similar contacts between the phosphate and ribose moieties and residues of the UCP1 cavity but not with the base moiety of UDP (Figure 7). UDP does not bind E191, W281, or Q85 but preferentially binds N184 and N188, which have little interaction with purine nucleotides. **Thus, these three residues might be responsible for the selectivity of UCP1 inhibition by purine nucleotides.**

In simulations, UDP and GDP show similar contacts between the phosphate and ribose moieties and residues of the UCP1 cavity but not with the base moiety of UDP (Figure 7). UDP does not bind E191, W281, or Q85 but preferentially binds N184 and N188, which have little interaction with purine nucleotides. **However, GTP rarely binds Q85 (Figure S1c), thus E191 and W281 seem to be the main residues responsible for the nucleotide selectivity.**

The Material & Methods section has been modified to include simulations with GTP and respiration measurements with GTP and palmitic acid.

3 - The authors described the interaction between GDP and phospholipids (lines 190-191). It would be interesting to study whether the FA simulations also describe an interaction between UCP1 and phospholipids when FAs are present. It would also be interesting to study whether phospholipids position themselves in the FA binding site when they are absent.

We have amended the text to make it clearer that we only see phospholipid binding when GDP is present, not for apo UCP1. As mentioned above, the binding site of FAs to UCP1 is still unknown, therefore, it is difficult to correlate this observation with fatty acid binding.

Line 204

Of note, we did not observe such intrusions of the phospholipids inside UCP1 cavity in the apo simulations.

4 - Mitochondrial respiration experiments were performed with lauric acid, a medium chain fatty acid. UCP1 shows stronger interactions with long chain fatty acids such as palmitic acid or arachidonic acid. It would be important to repeat these experiments, or at least key experiments with long chain fatty acids to rule out the possibility of inadequate interaction of lauric acid with UCP1.

We agree with the Reviewer that long chain fatty acids have been shown to better activate UCP1 than medium chain FA. We have performed respiration experiments using activation by palmitic acid. The results are reported in Supplementary Figure S9, and described in the text as mentioned in response to item 2 above.

Minor issues:

1 - It is stated in the text that AAC has been successfully used to simulate another uncoupling protein structure, UCP2. The Reviewer understands that this provides a rationale for using the AAC structure for another uncoupling protein, here UCP1. However, several works argue against UCP2 being a protein capable of transporting H⁺. Thus, successfully simulating UCP2 in the concept that it can transport H⁺ may not be a benchmark. It is important to rephrase this point (line 83). As set in the text, the authors performed simulations based on AACs not only because the structures of AACs are well established but also because only UCP1 and AACs have been confirmed to transport H⁺ in the inner mitochondrial membrane.

We agree with this remark. We have modified the sentence in question.

Line 75

High-resolution crystal structures exist for the ATP/ADP Carrier (AAC), which has also been shown to exhibit proton transport activity and to contribute to fatty-acid-dependent uncoupling of respiration [15]. Recently, AAC structures have been **successfully** used to model UCP2 [22].

High-resolution crystal structures exist for the ATP/ADP Carrier (AAC), which has also been shown to exhibit proton transport activity and to contribute to fatty-acid-dependent uncoupling of respiration [15]. Recently, AAC structures have been used to model UCP2 [22].

2 - It says line 113 "consistent with experimental evidence that UCP1 is accessible to ATP on both sides". The cited article is based on lipid bilayer experiments that could produce questionable results. Please expand and add references using alternative experiments.

We agree that the literature is ambiguous about this point. Since we have explored nucleotide entry through both sides to minimize the biases of the simulation, we have decided to explain our approach more precisely without references. Accordingly, we have modified this statement as follows:

Line 120

We explored binding to the cavity in both the C-state and the M-state models, in agreement with experimental evidence that UCP1 is accessible to ATP on both sides.

To avoid making assumptions about nucleotide access through the cytosol or the matrix, we explored binding to the cavity in both the C-state and the M-state models.

Reviewer #1 (Remarks to the Author):

The authors have taken our and other reviewer's comments into account and have modified the manuscript accordingly, making it more solid and consistent with the literature.

Reviewer #2 (Remarks to the Author):

On the whole my comments have been addressed by the authors. Personally I think the black sphere in figure 2 is too large and one could have managed with a smaller fiducial for the centre of the protein, without compromising the molecular details of the figure; but this is my personal opinion, and should not interfere with acceptance.

Reviewer #3 (Remarks to the Author):

The authors have satisfactorily addressed the reviewer's concerns.